# ASSUMPTION-LEAN INFERENCE ON TREATMENT EFFECT DISTRIBUTIONS

## ABSTRACT

In many fields, including healthcare, marketing, and online platform design, A/B tests are used to evaluate new treatments and make launch decisions based on average treatment effect (ATE) estimates. But this workflow can overlook distributional risks, such as a large fraction of individuals affected negatively by the treatment. Prior work in this setting has estimated partial identification bounds—known as Makarov bounds—on the cumulative distribution function of the treatment effect by making restrictive assumptions on the outcome distribution. In this paper, we propose a novel method for estimation and inference on Makarov bound that guarantees accurate estimation and valid asymptotic inference of the Makarov bounds for any outcome distribution under weaker assumptions. Our main technical contributions are to develop smoothed surrogates for the Makarov bounds, derive semiparametrically efficient estimators of these surrogates, and propose a procedure for optimal selection of the smoothing parameters. We show empirically on synthetic and semi-synthetic datasets that, by not relying on the assumptions made by other methods, our estimators achieve a better bias-variance trade-off and lower mean-squared error. Finally, we deploy our method on real A/B test data from a large social media platform, and show how estimates of the treatment effect distribution can inform decision-making.

## 1 INTRODUCTION

Randomized controlled trials (RCTs) and A/B tests are crucial for evaluating the impact of treatments or product changes on a target outcome. Examples include medical treatments aimed at improving patient health (Feuerriegel et al., 2024), advertising placement to increase revenue (Varian, 2016), and interventions on digital platforms to boost user engagement (Swaminathan & Joachims, 2015). By randomly assigning units to treatment and control, A/B tests enable unbiased comparisons of outcomes between groups. A common decision rule is to estimate the *average treatment effect* (ATE) and adopt the new treatment when the ATE is statistically significantly positive (Athey et al., 2020).

In many settings, the ATE is not sufficient for decision-making and practitioners in fact need estimates of distributional quantities beyond the average (Kallus & Zhou, 2021). In sensitive applications it is vital to understand the fraction of individuals that are affected negatively by the treatment, even when the average treatment effect is positive. For example, a minor tweak to a platform's ranking algorithm may lift overall engagement yet reduce engagement for new or low-activity users, increasing early churn. Such business-critical decisions motivate estimating the *entire distribution of the treatment effect*, rather than merely the mean.

Estimating the treatment effect distribution is challenging. Even in RCTs, where randomization rules out unobserved confounding, the joint distribution of potential outcomes is not identified (Fan & Park, 2010). Consequently, the treatment-effect distribution is only *partially* identified: its cumulative distribution function (c.d.f. ) is bounded by the sharp *Makarov bounds* (Makarov, 1982). These bounds are obtained by optimizing over all possible joint distributions of treatment and control that are consistent with the the observed marginal distributions of treatment and control. Hence the Makarov bounds are complex functionals of the data-generating distribution that involve suprema and infima over the outcome space. The non-smoothness of these suprema and infima complicates estimation of the bounds and especially complicates the application of standard tools from efficiency theory, such as debiased estimation (Kennedy, 2022).

Existing methods for efficient statistical inference on Makarov bounds either address stylized settings, such as a binary outcome (Kallus et al., 2022; Zhang & Richardson, 2025), or rely on a *margin assumption* (Semenova, 2025). The margin assumption requires the suprema and infima in the Makarov bounds to be attained at unique points, and is often violated in practice. For example, if the treatment effect is constant or nearly constant for a subset of users, the margin assumption will fail or nearly fail to hold. This naturally occurs when outcomes are discrete or exhibit zero inflation, which is a common occurrence in applied work. In this case, existing estimators are biased or high-variance, leading to inaccurate estimates and sub-optimal decisions.

In this paper, we propose a novel, assumption-lean, method for inference on Makarov bounds for the treatment-effect distribution. Our key idea is to replace the non-smooth suprema and infima that define the bounds with smooth surrogates, allowing us to apply semiparametric efficiency theory to obtain debiased estimators. We then derive an upper bound on the bias introduced by smoothing and incorporate it into both the selection of the smoothing parameters and the construction of our confidence intervals. Our method offers three key advantages: (i) it provides lower mean-squared error (MSE) estimates of Makarov bounds under violation of the margin assumption; (ii) it is model-agnostic and can be combined with arbitrary black-box learners to exploit covariates and tighten the bounds; and (iii) it admits data-driven procedures for selecting the smoothing level optimally.

**Our contributions are**[1]: (i) We propose debiased estimators of smoothed versions of the Makarov bounds and quantify the induced smoothing bias; (ii) We propose a new data-driven procedure to tune the smoothing parameters of our method; and (iii) We validate the effectiveness of our method using synthetic, semi-synthetic, and real-world data.

## 2 RELATED WORK

**Quantile treatment effects.** One stream of literature focuses on estimating quantile treatment effects (QTEs) (Abadie & Angrist, 1998; Chernozhukov & Hansen, 2005; Firpo, 2007). These are contrasts of quantiles of the potential outcomes distributions, defined via

$$\text{QTE}(\tau) \;=\; F_{Y(1)}^{-1}(\tau) - F_{Y(0)}^{-1}(\tau),$$

which compares the $\tau$-quantile of $Y(1)$ to the $\tau$-quantile of $Y(0)$. Note that QTEs do not quantify quantiles of the treatment effect distribution: QTEs describe how treatment shifts the marginal distributions of potential outcomes across individuals (for example, how the $\tau$-quantile under treatment differs from the $\tau$-quantile under control), while the latter would characterize the distribution of individual-level causal effects $Y(1) - Y(0)$ in the population. In contrast to QTEs, the treatment effect distribution is generally not identifiable from observed data without additional assumptions, and is the focus of our paper.

**Distributions of treatment effects.** There are two main streams of work on statistical inference for treatment-effect distributions. The first stream aims at constructing *prediction intervals* for the individual treatment effect (ITE), typically based on conformal prediction (Lei & Candès, 2021; Alaa et al., 2023; Schröder et al., 2024). These methods provide valid predictive intervals for individual-level effects, but do not allow for inference on the full treatment-effect distribution (e.g., estimating its c.d.f. or density).

The second stream focuses on inference on the c.d.f. of the treatment-effect distribution. However, this distribution is not identifiable under standard assumptions (Rubin, 1974; Robins, 1986; Fan & Park, 2010), even with experimental data. Two approaches to overcome this obstacle have emerged. The first is to impose additional assumptions on the data-generating process to achieve point identification, as in Post & Van Den Heuvel (2025). Such assumptions, however, are unrealistic and untestable. The second is to take a partial identification approach and estimate upper and lower *bounds* on the cumulative distribution function. For example, sharp bounds and corresponding estimators have been proposed for binary outcomes (Kallus et al., 2022; Zhang & Richardson, 2025).

For general outcomes, sharp bounds are given by the Makarov bounds (Makarov, 1982). Estimation and inference methods for these bounds are mostly based on *plug-in* approaches (Fan & Park, 2010; Ruiz & Padilla, 2022; Fava, 2024; Lee, 2024; Cui & Han, 2023; Liang & Wu, 2025), which do

---

[1]Code: `https://anonymous.4open.science/r/SmoothedMakarovBounds-632B`.

not leverage semiparametric efficiency theory (Kennedy, 2022). The resulting estimators are not guaranteed to be asymptotically normal and confidence intervals are often constructed via invalid bootstrap procedures. Beyond Makarov bounds, Firpo & Ridder (2019) introduced uniformly sharp bounds for the treatment-effect distribution, but these are computationally challenging as no closed-form expressions are available. Similarly, Ji et al. (2023) proposed a dual-optimization framework for partial identification, but this framework can yield overly conservative bounds when strong duality fails (see their Sec. 6.2).

**Efficient estimation of non-smooth causal estimands.** A key technical difficulty in the estimation of the Makarov bounds

Table 1: Approaches for debiased estimation of Makarov-bounds and other non-smooth causal estimands.

|  | Other causal estimands | Makarov bounds |
|---|---|---|
| Margin assumption | Policy value (Luedtke & Van Der Laan, 2016); Intersection bounds (Semenova, 2025); | (Melnychuk et al., 2024) (Semenova, 2025) |
| Smoothing | Optimal policy value (Park, 2024); IV bounds (Levis et al., 2025). | *Ours.* |

is that the bounds involve suprema and infima over the outcome space. These suprema and infima are non-differentiable, precluding the use of standard theory.

One literature stream attempts to remedy this by making a so-called *margin assumption* (Kitagawa & Tetenov, 2018), which essentially requires that the suprema and infima are attained at a single point only. This assumption has been used in the estimation of various other statistical estimands with similar non-differentiability issues: the optimal policy value (Luedtke & Van Der Laan, 2016), covariate-conditional Makarov bounds (Melnychuk et al., 2024), and more general intersection bounds (Semenova, 2025) which include Makarov bounds as a special case. However, the margin assumption is violated even in simple settings, like that of a constant treatment effect, making these methods unreliable in practice.

A second stream of work handles non-differentiability by approximating non-differentiable functions (e.g., the supremum) with smooth functions. This approach has been developed for inference on the optimal policy value Park (2024) and on bounds on average treatment effects in instrumental variable settings Levis et al. (2025). To the best of our knowledge no prior work has proposed a similar method for estimation and inference of Makarov bounds, which is the scope of our paper.

## 3  PROBLEM SETUP

### 3.1  SETTING

**Data:** We consider a causal inference setting using either randomized or observational data. That is, we consider a population $Z = (X, A, Y) \sim \mathbb{P}$, where $X \in \mathcal{X} \subseteq \mathbb{R}^d$ are covariates, $A \in \{0, 1\}$ is a binary action, and $Y \in \mathbb{R}$ is a continuous outcome of interest that is observed after taking the action $A$. For example, $X$ may be user demographics, $A$ may be a binary decision of whether a policy is implemented, and $Y$ may be a user engagement metric. We provide extensions of all our results to discrete outcomes in Appendix C. We also assume that we have access to a dataset $\mathcal{D} = \{(x_i, a_i, y_i)\}_{i=1}^n$ of size $n \in \mathbb{N}$ sampled i.i.d. from $\mathbb{P}$.

**Notation.** We define the *propensity score* as $\pi(x) = \mathbb{P}(A = 1 \mid X = x)$. The propensity score characterizes the treatment assignment mechanism and is often known in randomized experiments, e.g., $\pi(x) = 0.5$. Furthermore, we define the *response distributions* as the conditional outcome c.d.f.s $F_a(y|x) = \mathbb{P}(Y \leq y | X = x, A = a)$ for $a \in 0, 1$. We let $\eta = \{\pi, F_1, F_0\}$ be the collection of these *nuisance functions*. For any $v \in \mathbb{R}$, we write $v_+ = \max(v, 0)$ and $v_- = \min(v, 0)$.

**Target estimand.** We use the potential outcomes framework of Rubin (1974) and denote $Y(a)$ as the potential outcome corresponding to the treatment $A = a$. The parameter of interest is the cumulative distribution function of the treatment effect evaluated at a single point,

$$\rho(\delta) = \mathbb{P}(Y(1) - Y(0) \leq \delta). \tag{1}$$

The c.d.f. $\rho(\delta)$ characterizes the entire treatment effect distribution. For example, $\rho(0)$ is the fraction of users negatively affected by the treatment (Kallus et al., 2022).

**Partial identification bounds.** We impose the following standard causal inference assumptions (Rubin, 1974).

**Assumption 3.1** (Standard causal inference assumptions)**.** For all $a \in \{0, 1\}$ and $x \in \mathcal{X}$ we have: (i) *consistency*, $Y(a) = Y$ whenever $A = a$; (ii) *overlap*, $0 < \pi(x) < 1$ whenever $\mathbb{P}(X = x) > 0$; and (iii) *ignorability*, $A \perp Y(1), Y(0) \mid X = x$.

Conditions (ii) and (iii) are automatically satisfied in randomized experiments where the propensity score $\pi$ is known and can be controlled (and is often the constant function $\pi(x) = 0.5$). Condition (i) prohibits interference between individuals and excludes the possibility of spillover effects. Together, these assumptions allow us to *partially identify* the treatment effect distribution $\rho(\delta)$ via the Makarov bounds (Makarov, 1982)

$$\rho^-(\delta) \le \rho(\delta) \le \rho^+(\delta), \tag{2}$$

where

$$\rho^-(\delta) = \mathbb{E}\left[\sup_{y \in \mathcal{Y}} \left(F_1(y|X) - F_0(y - \delta|X)\right)_+\right] \text{ and } \rho^+(\delta) = 1 + \mathbb{E}\left[\inf_{y \in \mathcal{Y}} \left(F_1(y|X) - F_0(y - \delta|X)\right)_-\right]. \tag{3}$$

The Makarov bounds $\rho^-(\delta)$ and $\rho^+(\delta)$ only depend on the marginal response distributions $F_1$ and $F_0$ and are thus identified from the observation distribution $\mathbb{P}$. Intuitively, the Makarov bounds quantify the stochastically smallest and largest distributions that can be obtained by maximizing or minimizing over joint distributions of the potential outcomes that are compatible with the observed marginals $F_1$ and $F_0$. (They are also generalizations of the bounds from (Kallus, 2022) for the fraction of negatively affected users in the case of binary outcomes.)

## 3.2 BACKGROUND ON ESTIMATING MAKAROV BOUNDS

**Plug-in estimation.** The simplest estimator of the Makarov bounds is the so-called *plug-in estimator* (Fan & Park, 2010): one first obtains nuisance estimators $\hat{\eta} = (\hat{F}_1, \hat{F}_0, \hat{\pi})$ of the response distributions $F_a$ and propensity score $\pi$ (in the observational setting). Then, one substitutes the estimated nuisance functions into the expression for the bound from Eq. (3) to obtain

$$\hat{\rho}_{\mathrm{PI}}^-(\delta) = \frac{1}{n} \sum_{i=1}^n \sup_{y \in \mathcal{Y}} \left(\hat{F}_1(y|x_i) - \hat{F}_0(y - \delta|x_i)\right)_+. \tag{4}$$

However, it turns out that the plugin estimator $\hat{\rho}_{\mathrm{PI}}^-(\delta)$ has two drawbacks (Kennedy, 2022): (i) plug-in bias, which makes it asymptotically suboptimal, and (ii) difficulty establishing asymptotic normality under reasonable conditions on the nuisance estimators, which prevents reliable confidence interval construction.

**Semiparametric efficient estimators.** To address these limitations, a major line of work in causal inference leverages *semiparametric efficiency theory* to construct *debiased estimators*. Debiased estimators are of the form

$$\hat{\rho}_{\mathrm{AIPTW}}^-(\delta) = \hat{\rho}_{\mathrm{PI}}^-(\delta) + \frac{1}{n} \sum_{i=1}^n \Psi_{\delta, y^*}^- \left(z_i, \hat{\eta}, \hat{\rho}_{\mathrm{PI}}^-(\delta)\right), \tag{5}$$

where $\Psi_{\delta, y^*}^- \left(Z, \eta, \rho^-(\delta)\right)$ is the so-called *efficient influence function* (EIF) of the target estimand. Adding the EIF term debiases the plug-in estimator, yielding an estimator that is asymptotically efficient and normally distributed, i.e., yielding the lowest possible asymptotic variance given by $\mathbb{E}[\Psi_{\delta, y^*}^- \left(Z, \eta, \rho^-(\delta)\right)^2]$. Hence, debiasing using the EIF enables valid and efficient statistical inference.

**Envelope estimators for Makarov bounds.** For the lower Makarov bound, the efficient influence function is given by (Melnychuk et al., 2024; Semenova, 2025):

$$\Psi_{\delta, y^*}^- \left(Z, \eta, \rho^-(\delta)\right) = d_{\delta, \eta}(y^*|X) - \rho^-(\delta) + \mathbf{1}(d_{\delta, \eta}(y^*|X) > 0)\left(\frac{A}{\pi(X)} \left(\mathbf{1}(Y \le y^*) - F_1(y^*|X)\right) \right. \tag{6}$$

$$\left. - \frac{1 - A}{1 - \pi(X)} \left(\mathbf{1}(Y \le y^* - \delta) - F_0(y^* - \delta|X)\right)\right), \tag{7}$$

where $d_{\delta,\eta}(y|X) = F_1(y|X) - F_0(y - \delta|X)$ and $y^* \in \arg\max_{y \in \mathcal{Y}} d_{\delta,\eta}(y|X)$.

For each observation $x_i$, $y^*$ can be computed (e.g., via grid search) and plugged into the debiased estimator from Eq. (5). This approach is known as an *envelope estimator* (Semenova, 2025).

**Margin assumption.** Debiased estimators rely on several assumptions, including pathwise differentiability (van der Laan & Rubin, 2006). For Makarov bounds, this essentially translates to requiring the supremum and infimum $y^*$ to yield unique solutions, which is known as the *margin assumption* (Kitagawa & Tetenov, 2018). However, this assumption is violated even for the simple case of constant treatment effects for a subset of users. Indeed, if the treatment effect is constant for users with some values of $X$, then the treatment c.d.f. $F_1(y \mid X)$ for these users is a constant vertical shift of the control c.d.f. $F_0(y \mid X)$, and the supremum/ infimum $y^*$ can be attained at many values of $y$ (see Fig. 1 for an illustration). Importantly, such (near-) constant effects are not just a feature of stylized toy examples. They naturally occur when outcomes are discrete or exhibit zero inflation, which is a common occurrence in applied work. For instance, in A/B tests with binary or highly skewed outcomes such as clicks, conversions, or purchases (many units with exactly zero events). Thus, the margin assumption fails and the envelope estimator is no longer efficient. This inefficiency can manifest a sub-optimal mean-squared-error, because the margin assumption can inflate the variance of the estimator which can lead to suboptimal bias-variance trade-off as we will show later.

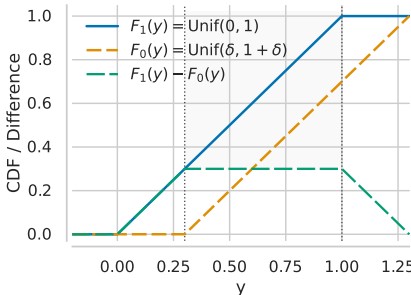

Figure 1: **Example for violating the margin assumption.** Depicted are two shifted (uniform) c.d.f.s that violate the margin assumption, leading to a plateau where the argmax in not unique.

We emphasize that the failure of the margin assumption is not restricted to pathological edge cases. Constant treatment effects are a basic model of causal inference, and since effect sizes are often extremely small in digital experiments, we are often very nearly in the setting of a constant zero treatment effect. We cannot confidently use estimators that rely on a margin assumption when analyzing real-world data, necessitating the development of new methods.

## 4 ESTIMATING MAKAROV BOUNDS IN THE PRESENCE OF MARGIN VIOLATIONS

In this section, we present our methodology for valid inference on Makarov bounds when the margin assumption is violated. Our strategy proceeds in three steps. First, in Sec.4.1, we introduce a smooth surrogate approximation of the Makarov bounds that is amenable to debiased estimation. Next, in Sec.4.2, we derive debiased estimators for these smoothed bounds and construct confidence intervals that achieve valid coverage. Finally, in Sec. 4.3, we propose a data-driven procedure to select the smoothing parameters in a principled way.

### 4.1 A SMOOTH APPROXIMATION FOR THE MAKAROV BOUNDS

Two key components of the Makarov bounds in Eq. (3) prevent debiased inference in the presence of margin violations: (i) the supremum and infumum operators, $\sup_{y \in \mathcal{Y}}$ and $\inf_{y \in \mathcal{Y}}$, and (ii) the positivity and negativity operators, $(\cdot)_+ = \max(\cdot, 0)$ and $(\cdot)_- = \min(\cdot, 0)$. Both components are non-smooth in the presence of margin violations. Intuitively, when the optimizer has multiple solutions, even small estimation errors in the nuisance functions can lead to large and unstable variation in the estimator, as well as bias in the wrong direction. This instability is what invalidates asymptotic efficiency and reliable confidence intervals.

**Bound approximations**. Our key idea is to replace these problematic components with smooth approximations that return unique values, even when the original functions are flat or multi-valued. This smoothing restores pathwise differentiability, enabling debiased estimation and valid inference without requiring the margin assumption. For the suprema and infima, we adapt the approximation from (Levis et al., 2025) for discrete maxima to continuous variables and define the continuous

log-sum-exp operator (LSE) $g_{t_1} : \mathcal{F} \to \mathbb{R}$ via

$$g_{t_1}(f) = \frac{1}{t_1} \log \left( \int \exp(t_1 f(y)) \, dy \right) \tag{8}$$

For (ii), we replace $v_+$ and $v_-$ with the softplus function $h_{t_2} : \mathbb{R} \to \mathbb{R}$ via $h_{t_2}(u) = \frac{1}{t_2} \log\left(1 + e^{t_2 u}\right)$. The following lemma shows that replacing the components in (i) and (ii) with $g_{t_1}$ and $h_{t_2}$ gives good approximations of the Makarov bounds.

**Lemma 4.1** (Makarov bound approximations). *Assume that $\mathcal{Y}$ is compact with finite Lebesque measure $|\mathcal{Y}|$. For any $t_1, t_2 > 0$, we define the smoothed Makarov bounds via*

$$\rho_{t_1,t_2}^-(\delta) = \mathbb{E}\left[\frac{1}{t_2} \log\left(1 + I_{t_1,\delta}(X)^{\frac{t_2}{t_1}}\right)\right] \text{ and } \rho_{t_1,t_2}^+(\delta) = 1 - \mathbb{E}\left[\frac{1}{t_2} \log\left(1 + I_{-t_1,\delta}(X)^{\frac{t_2}{t_1}}\right)\right], \tag{9}$$

*where*

$$I_{t,\delta}(x) = \int_{\mathcal{Y}} \exp\bigl(t[F_1(y|x) - F_0(y - \delta|x)]\bigr) \, dy. \tag{10}$$

*Then, it holds that*

$$\rho_{t_1,t_2}^-(\delta) - b(t_1,t_2) \leq \rho^-(\delta) \text{ and } \rho^+(\delta) \leq \rho_{t_1,t_2}^+(\delta) + b(t_1,t_2), \tag{11}$$

*where $b(t_1,t_2) = \frac{\log(2)}{t_2} + \frac{(\log|\mathcal{Y}|)_+}{t_1}$ quantifies the approximation bias. Furthermore, $\rho_{t_1,t_2}^-(\delta) \to \rho^-(\delta)$ and $\rho_{t_1,t_2}^+(\delta) \to \rho^+(\delta)$ as $t_1, t_2 \to \infty$.*

*Proof.* See Appendix A. $\qquad\square$

Intuitively, this smoothing replaces the "hard max" operation of the supremum with a "soft max." Instead of arbitrarily selecting one of multiple equally optimal points, the approximation takes a smooth weighted average that changes gradually as the nuisance estimates shift. This makes the estimator more stable and allows for establishing asymptotic normality of debiased estimators.

## 4.2 DEBIASED ESTIMATORS

We now derive the debiased estimators for the smoothed bounds in Lemma 4.1. The following result establishes the efficient influence functions (EIF) of the lower and upper approximation.

**Theorem 4.2.** *The efficient influence functions of the smoothed Makarov bounds $\rho_{t_1,t_2}^-(\delta)$ and $\rho_{t_1,t_2}^+(\delta)$ are given by*

$$\Psi_{t_1,t_2,\delta}^-\left(Z, \eta, \rho_{t_1,t_2}^-(\delta)\right) = \sigma_{t_1,t_2,\delta}^-(X)\left[\frac{A}{\pi(X)}\Phi_{t_1,\delta}^1(X,Y) - \frac{1-A}{1-\pi(X)}\Phi_{t_1,\delta}^0(X,Y)\right] \tag{12}$$

$$+ \frac{1}{t_2}\log\left(1 + I_{t_1,\delta}(X)^{\frac{t_2}{t_1}}\right) - \rho_{t_1,t_2}^-(\delta), \tag{13}$$

$$\Psi_{t_1,t_2,\delta}^+\left(Z, \eta, \rho_{t_1,t_2}^+(\delta)\right) = \sigma_{t_1,t_2,\delta}^+(X)\left[\frac{A}{\pi(X)}\Phi_{-t_1,\delta}^1(X,Y) - \frac{1-A}{1-\pi(X)}\Phi_{-t_1,\delta}^0(X,Y)\right] \tag{14}$$

$$+ 1 - \frac{1}{t_2}\log\left(1 + I_{-t_1,\delta}(X)^{\frac{t_2}{t_1}}\right) - \rho_{t_1,t_2}^+(\delta), \tag{15}$$

*where we denote*

$$\Phi_{t_1,\delta}^a(X,Y) = \int_{\mathcal{Y}} w_{t_1,\delta}(y \mid X)\left(\mathbf{1}\{Y \leq y - (1-a)\delta\} - F_a(y - (1-a)\delta \mid X)\right) dy, \tag{16}$$

$$w_{t_1,\delta}(y|x) = \frac{\exp\bigl(t_1(F_1(y|x) - F_0(y - \delta|x))\bigr)}{I_{t_1,\delta}(x)}, \tag{17}$$

$$\sigma_{t_1,t_2,\delta}^-(x) = \frac{I_{t_1,\delta}(x)^{t_2/t_1}}{1 + I_{t_1,\delta}(x)^{t_2/t_1}}, \text{ and } \sigma_{t_1,t_2,\delta}^+(x) = \frac{I_{-t_1,\delta}(x)^{t_2/t_1}}{1 + I_{-t_1,\delta}(x)^{t_2/t_1}}. \tag{18}$$

*Proof.* See Appendix A. □

The smoothed EIFs from Theorem 4.2 are of similar structure as the envelope EIF from Eq. (6). Cruicially however, the terms involving the argmax/argmin $y^*$ are replaced by smooth weighted integrals $\Phi_{t_1,\delta}^a(X,Y)$, and the indicator $\mathbf{1}(d_{\delta,\eta}(y^*|X) > 0)$ is replaced with one of the smooth approximations $\sigma_{t_1,t_2,\delta}^{\pm}(X)$.

Using Theorem 4.2, we can now obtain debiased estimators for the smoothed Makarov bounds. Following Eq. (5), these are given by

$$\hat{\rho}_{t_1,t_2}^-(\delta) = \frac{1}{n} \sum_{i=1}^n \frac{1}{t_2} \log\left(1 + \hat{I}_{t_1,\delta}(x_i)^{\frac{t_2}{t_1}}\right) + \hat{\sigma}_{t_1,t_2,\delta}^-(x_i)\left[\frac{a_i - \hat{\pi}(x_i)}{\hat{\pi}(x_i)(1 - \hat{\pi}(x_i))}\,\hat{\Phi}_{t_1,\delta}^{a_i}(x_i,y_i)\right],$$

$$(19)$$

$$\hat{\rho}_{t_1,t_2}^+(\delta) = \frac{1}{n} \sum_{i=1}^n 1 - \frac{1}{t_2} \log\left(1 + \hat{I}_{-t_1,\delta}(x_i)^{\frac{t_2}{t_1}}\right) + \hat{\sigma}_{t_1,t_2,\delta}^+(x_i)\left[\frac{a_i - \hat{\pi}(x_i)}{\hat{\pi}(x_i)(1 - \hat{\pi}(x_i))}\,\hat{\Phi}_{-t_1,\delta}^{a_i}(x_i,y_i)\right].$$

**Correcting for approximation bias.** Using above debiased estimators enables efficient and asymptotically normal inference, and thus construction of confidence intervals for the smoothed Makarov bounds. However, it remains to translate these intervals to the non-smoothed Makarov bounds, as the smoothing may introduce bias. Luckily, we can leverage Lemma 4.1 to upper bound this approximation bias via the term $b(t_1, t_2)$. We can then add this term to the one-sided confidence intervals for the upper and lower Makarov bounds, yielding valid confidence intervals as follows.

**Corollary 4.3** (Asymptotic confidence interval for the treatment effect distribution)**.** *Assume the nuisance estimators $\hat{\eta}$ are obtained on an independent sample from $\mathcal{D}$ (e.g., via sample splitting or cross-fitting). Under standard regularity and rate conditions for orthogonal/debiased estimation (e.g., Chernozhukov et al., 2018), it holds for each fixed $\delta$ that*

$$\lim_{n\to\infty} \mathbb{P}\Big(c_{t_1,t_2,\delta,\alpha}^-(\mathcal{D}_n) \le \rho(\delta) \le c_{t_1,t_2,\delta,\alpha}^+(\mathcal{D}_n)\Big) \ge 1 - \alpha, \tag{20}$$

*where*

$$c_{t_1,t_2,\delta,\alpha}^{\pm}(\mathcal{D}_n) = \hat{\rho}_{t_1,t_2}^{\pm}(\delta) \pm z_{1-\alpha/2}\sqrt{\frac{1}{n^2}\sum_{i=1}^n\Big(\Psi_{t_1,t_2,\delta}^{\pm}\big(z_i,\hat{\eta},\hat{\rho}_{t_1,t_2}^{\pm}(\delta)\big)\Big)^2} \pm b(t_1,t_2), \tag{21}$$

$b(t_1, t_2)$ *is from Lemma 4.1,* $\Psi_{t_1,t_2,\delta}^{\pm}$ *are given in Theorem 4.2, and* $z_{1-\frac{\alpha}{2}}$ *denotes the* $1 - \frac{\alpha}{2}$ *standard normal quantile.*

*Proof.* See Appendix A. □

## 4.3 Smoothing parameter selection

**Bias-variance trade-off.** Our smoothed debiased estimators from Eq. (19) depend on the smoothing parameters $t_1$ and $t_2$ that quantify the approximation quality of the non-smooth bound components. This implies a trade-off between the approximation-induced bias and the variance reduction under margin violation. Formally, a standard MSE decomposition yields

$$\mathbb{E}_n\big[(\hat{\rho}_{t_1,t_2}^-(\delta) - \rho^-(\delta))^2\big] = \underbrace{\big(\rho_{t_1,t_2}^-(\delta) - \rho^-(\delta)\big)^2}_{\text{approximation (smoothing) bias}^2} + \underbrace{\frac{1}{n}\mathbb{E}\big[\Psi_{t_1,t_2,\delta}^-\big(Z,\eta,\rho_{t_1,t_2}^-(\delta)\big)^2\big]}_{\text{asymptotic variance}/n} + \underbrace{o\Big(\frac{1}{n}\Big)}_{\text{remainder}}. \tag{22}$$

As a consequence, choosing the smoothing parameters $t_1$ and $t_2$ correctly is crucial to obtain a low MSE that optimizes the bias-variance trade-off. We propose two different methods for data-driven smoothing-parameter selection: (i) minimizing an upper bound on the MSE, and (ii) using Lepski's method. Our full proposed procedure is shown in Algorithm 1.

**Algorithm 1:** Smoothed estimation of Makarov bounds with sample splitting.

1: **Input:** Dataset $\mathcal{D} = \{(x_i, a_i, y_i)\}_{i=1}^n$
2: **Stage 0:** Split the data $\mathcal{D}$ randomly into two disjoint datasets $\mathcal{D}_1$ and $\mathcal{D}_2$.
3: **Stage 1a:** Estimate nuisance functions $\hat{\eta} = (\hat{F}_1, \hat{F}_0, \hat{\pi})$ on $\mathcal{D}_1$; obtain predictions on $\mathcal{D}_2$.
4: **Stage 1b:** Obtain tuned smoothing parameters $\hat{t}_1^{\pm}$ and $\hat{t}_2^{\pm}$ on $\mathcal{D}_1$.
5: **Stage 2:** Compute debiased estimators $\hat{\rho}_{\hat{t}_1^-, \hat{t}_2^-}^-(\delta)$ and $\hat{\rho}_{\hat{t}_1^+, \hat{t}_2^+}^+(\delta)$.
6: **Output:** $g_\theta$ and $\alpha_\phi$.

**Minimizing an upper bound on the MSE.** Using Lemma 4.1, we can replace the smoothing bias term in Eq. (22) with $b(t_1, t_2) = \frac{\log(2)}{t_2} + \frac{(\log|\mathcal{Y}|)_+}{t_1}$, which yields the data-driven tuning rule $\arg\min_{t_1, t_2} b(t_1, t_2)^2 + \frac{1}{n} \sum_{i=1}^n \Psi_{t_1, t_2, \delta}^-(z_i, \hat{\eta}, \hat{\rho}_{t_1, t_2}^-(\delta))^2$, which can be minimized via grid search.

**Lepski's method.** Lepski's method (Lepskii, 1992; 1993) selects the smallest amount of smoothing whose estimate is statistically indistinguishable from estimates obtained with *less* smoothing, thereby controlling approximation bias without paying unnecessary variance. We start by ranking all candidate pairs $(t_1^{(k)}, t_2^{(k)})$ within the grid by the size of their associated bias term $B_k = b(t_1^{(k)}, t_2^{(k)})$ in *descending* order (more smoothing first). Scanning in this order, for each $k$ compare the corresponding estimates $\hat{\rho}_{t_1^{(k)}, t_2^{(k)}}^-(\delta)$ to a small set of less–smoothed candidates $r$ with $B_r < B_k$ and accept the first $k$ such that $\left| \hat{\rho}_{t_1^{(k)}, t_2^{(k)}}^-(\delta) - \hat{\rho}_{t_1^{(r)}, t_2^{(r)}}^-(\delta) \right| \leq z_{1-\alpha/2} \widehat{\mathrm{SE}}(k, r)$ for all such $r$, where the tolerance uses EIF differences $\widehat{\mathrm{SE}}(k, r) = 1/\sqrt{n} \, \mathrm{sd}_n\left( \widehat{\Psi}_{t_1^{(k)}, t_2^{(k)}, \delta}^-(Z_i, \hat{\eta}, \hat{\rho}_{t_1^{(k)}, t_2^{(k)}}^-(\delta)) - \widehat{\Psi}_{t_1^{(r)}, t_2^{(r)}, \delta}^-(Z_i, \hat{\eta}, \hat{\rho}_{t_1^{(r)}, t_2^{(r)}}^-(\delta)) \right)$, and $\mathrm{sd}_n(\cdot)$ is the empirical standard deviation over $i = 1, \ldots, n$.

## 5 EXPERIMENTS

We now confirm the effectiveness of our proposed method empirically. As is standard in causal inference (Shalit et al., 2017; Curth & van der Schaar, 2021), we evaluate our method on synthetic and semi-synthetic data where we have access to ground-truth values of causal quantities. We also provide experimental results using real-world A/B tests. Additional experimental results are in Appendix I.

**Nuisance estimation.** We estimate the treatment-specific conditional c.d.f.s $F_a(\cdot \mid x)$, $a \in \{0, 1\}$, via likelihood-based gradient boosting with sample-splitting. For continuous outcomes we fit covariate-dependent Gaussian mixtures; for discrete outcomes we use a multinomial classifier and obtain the c.d.f. by cumulatively summing predicted class probabilities; and for nonnegative counts with excess zeros we employ a zero-inflated Poisson with covariate-dependent rate and zero-inflation. Training uses early stopping on held-out log-likelihood. Each learner outputs $\hat{F}_a(y \mid x)$ on an arbitrary grid in $\mathcal{Y}$ of size $k = 300$ that is used for numerical integration. Details are in Appendix F.

**Baselines.** We compare our method with the *plug-in estimator* from Eq.(4) and the *envelope estimator* from Eq. (6). To ensure a fair comparison, we use the same nuisance estimators for $F_1(y|x)$ and $F_0(y|x)$ from above. Additionally, we also report results for our method and the baselines targeting the *marginal* Makarov bounds (corresponding to not using any covariates $X$). For the marginal bounds, we use the standard empricial c.d.f. on a holdout dataset as a nuisance estimator. We use Lepski's method for selecting the smoothing parameters.

**Evaluation.** For synthetic data with available ground-truth we report the estimation error $|\rho^{\pm}(\delta) - \hat{\rho}^{\pm}(\delta)|$ averaged over $K = 2500$ Monte Carlo runs for both the lower and the upper Makarov bound. For real-world data without ground-truth we report the estimated bounds and confidence intervals.

### 5.1 (SEMI)-SYNTHETIC DATA

**Fully synthetic data.** We simulate data from a synthetic data-generating process (DGP) that violates the margin assumption by having a constant treatment effect for some users. This leads to plateaus like those in Fig.1 in the difference between the treatment and control c.d.f. (see Appendix G).

We consider two variations of this DGP: (i) a variation with a smaller plateau width corresponding to only small assumption violation, and (ii) a variation with a larger plateau width corresponding to larger assumption violation. A comparison of our method to existing methods for both of these DGPs, for both the upper and lower bounds, with and without covariates, is given in Table. 2 (top)

| Data | Bound | Side | Small assumption violation | | | Large assumption violation | | |
|---|---|---|---|---|---|---|---|---|
| | | | Plugin | Envelope | Ours | Plugin | Envelope | Ours |
| Synthetic | Marginal | lower | $3.98 \pm 0.11$ | $2.00 \pm 0.05$ | $\mathbf{1.81 \pm 0.04}$ | $5.59 \pm 0.11$ | $1.35 \pm 0.08$ | $\mathbf{0.82 \pm 0.06}$ |
| | | upper | $4.53 \pm 0.11$ | $1.99 \pm 0.06$ | $\mathbf{1.92 \pm 0.05}$ | $5.55 \pm 0.11$ | $1.40 \pm 0.08$ | $\mathbf{0.83 \pm 0.06}$ |
| | Cov.-adjusted | lower | $5.02 \pm 0.12$ | $1.86 \pm 0.05$ | $\mathbf{1.81 \pm 0.04}$ | $6.42 \pm 0.13$ | $1.28 \pm 0.07$ | $\mathbf{0.90 \pm 0.06}$ |
| | | upper | $4.67 \pm 0.11$ | $1.93 \pm 0.05$ | $\mathbf{1.84 \pm 0.04}$ | $6.29 \pm 0.12$ | $1.25 \pm 0.07$ | $\mathbf{0.86 \pm 0.06}$ |
| OHIE | Marginal | lower | $1.95 \pm 0.06$ | $1.21 \pm 0.03$ | $\mathbf{0.76 \pm 0.02}$ | $2.22 \pm 0.06$ | $1.14 \pm 0.03$ | $\mathbf{0.77 \pm 0.02}$ |
| | | upper | $2.40 \pm 0.07$ | $1.46 \pm 0.04$ | $\mathbf{1.09 \pm 0.03}$ | $2.52 \pm 0.07$ | $1.31 \pm 0.04$ | $\mathbf{0.97 \pm 0.03}$ |
| | Cov.-adjusted | lower | $4.88 \pm 0.09$ | $1.46 \pm 0.03$ | $\mathbf{1.44 \pm 0.03}$ | $4.71 \pm 0.09$ | $1.09 \pm 0.03$ | $\mathbf{1.00 \pm 0.02}$ |
| | | upper | $5.29 \pm 0.09$ | $1.50 \pm 0.04$ | $\mathbf{1.41 \pm 0.03}$ | $4.31 \pm 0.09$ | $1.28 \pm 0.03$ | $\mathbf{1.08 \pm 0.03}$ |

Table 2: Average mean-squared error of estimators of $\rho(0)$ across settings. Error bars are 95% CIs across 2500 replications. Our method consistently attains the lowest mean-squared error.

and shows that our **method achieves the best estimation performance** across all bound types and settings. Furthermore, the gap between our estimator and the baselines widens under stronger margin violation, highlighting that our method efficiently handles these violations when the baselines do not.

**Semi-synthetic data.** For our semi-synthetic experiments, we use covariate data from the *Oregon health insurance experiment* (OHIE) (Finkelstein et al., 2012). The OHIE was a randomized experiment meant to assess the effect of health insurance on outcomes such as health or economic status (see Appendix H for details). We use the following covariates: age, gender, language, the number of emergency visits before the experiment, and the number of people the individual signed up with. We then simulate treatment and outcomes such that the margin assumption is violated. Again, we consider two different settings with different the severities of assumption violation. The results are shown in Table. 2 (bottom). Again, **our method achieves the best estimation performance** across all bound types and settings and the gap between the smoothing and envelope estimators widens when the degree of assumption violation increases.

## 5.2 REAL-WORLD DATA

**A/B tests from a consumer technology company.** Finally, we apply our method to three real A/B tests from a large consumer technology company. All three A/B tests have constant propensity score $\pi(x) = 0.5$ and have on the order of tens of millions of observations. The first experiment is an experiment that boosts certain content in a ranking context, and the outcome metric is a measure of engagement. The second experiment demotes certain content in a (different) ranking context, and the outcome metric is a (different) measure of engagement. The third experiment tests a treatment meant to increase visitation, and the outcome metric is a visitation metric. Basic statistics for these experiments are reported in Table 3. For each experiment, we estimate the Makarov bounds to obtain a partial identification region for the c.d.f. of the treatment effect.

Table 3: Experiment statistics summary.

| | Sample size, $n$ | % ATE | Upper bound on $\rho(-1)^{(*)}$ |
|---|---|---|---|
| Exp. 1 | $\approx 30$m | 16.25% | 0.37 |
| Exp. 2 | $\approx 5$m | $-1.76\%$ | 0.76 |
| Exp. 3 | $\approx 10$m | 0.45% | 0.99 |

Sample sizes approximate millions (m). $(*)$ Setting $\delta = -1$ implies negative treatment effect.

*Results.* The results are shown in Figure 2. For the first experiment, we see that the partial identification region is relatively narrow. We are confident that $\mathbb{P}(Y_i(1) - Y_i(0) \le -1) = \mathbb{P}(Y_i(1) - Y_i(0) < 0)$ is no more than 0.25, meaning that there are very few users for whom the treatment is decreasing engagement. Since we are confident this treatment does not negatively affect a significant fraction of users, and the ATE estimate in Table 3 is positive, this treatment would be safe to launch. (On the other hand, the vertical jump in the identification region at 0 and the high lower bound on $\mathbb{P}(Y_i(1) - Y_i(0) \le 0)$ suggest that—despite the positive ATE estimate—the treatment is actually having no effect on most users, likely due to the dynamics of the ranking, meaning that we may want to search for more effective treatments.)

For the second experiment, the partial identification region is wider—our upper bound on $\mathbb{P}(Y_i(1) - Y_i(0) < 0)$ is now 0.74, suggesting that this treatment may in fact affect a majority of users negatively, and we should conduct further analysis before launching it. Finally, for the third experiment, the partial identification region is extremely wide: our upper bound on $\mathbb{P}(Y_i(1) - Y_i(0) < 0)$ is now 0.99, suggesting that this treatment could potentially have a negative effect on nearly all users, and would be inadvisable to launch. Note that this experiment has a positive average treatment effect, as seen in Table 3, and so this recommendation to not launch contradicts the standard decision-making process.

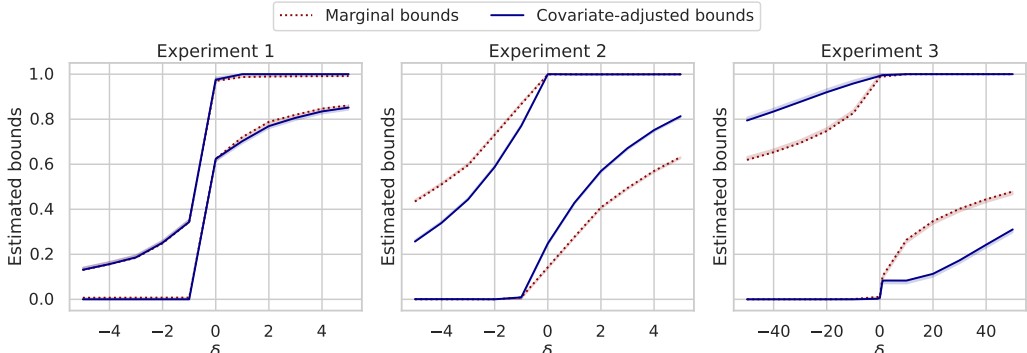

Figure 2: Estimated Makarov bounds for three A/B tests from a consumer technology company. Based on these bounds, we are confident that Experiment 1 does not negatively impact many users, and thus is safe to launch, whereas Experiments 2 and 3 may negatively impact many users.

Taken together, these three examples highlight how our methods can be used in real world settings to characterize treatment effect distributions more completely than the average treatment effect, and thus inform launch decisions. We also provide additional comparisons with baselines in Appendix I.

# 6 DISCUSSION.

In this paper, we proposed a new method for inferring Makarov bounds on the treatment effect distribution without making a margin assumption.

**Limitations and future work.** Our current approach is limited to static binary treatments. Future work may consider extending our approach to continuous or time-varying treatments and outcomes. Additionally, future work may also consider extensions to unbounded outcome spaces or settings with unit interference (common in A/B testing).

**Broader impact.** Our method enables practitioners to make inferences about treatment effect distribution without relying on untestable assumptions, thus improving the reliability of established methods.

**Ethics statement.** We adhere to the ICLR Code of Ethics and acknowledge this during submission. Our work analyzes synthetic, semi-synthetic, and anonymized A/B-test data; no personal data are released or re-identified. The methods aim to improve safety by quantifying the fraction potentially harmed, but misuse is possible; we therefore emphasize risk-aware reporting and recommend subgroup audits when sensitive attributes are used.

**Reproducibility Statement.** An anonymized repository (linked in the submission) provides code to train/evaluate our estimators, select smoothing parameters, and compute one-sided confidence intervals. The paper specifies assumptions and estimands (Problem Setup), the smoothing/bias bounds and EIF-based estimators (Method), and evaluation protocols (Experiments), with full proofs and additional details in the appendix. We release synthetic and semi-synthetic generators, document preprocessing, model classes, hyperparameters, seeds, and scripts to recreate all synthetic and semi-synthetic figures/tables. Proprietary A/B data cannot be shared, but we plan to release anonymized data and code upon acceptance.

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

## A PROOFS.

### A.1 PROOF OF LEMMA 4.1

*Proof.* Recall that the LSE operator is defined as

$$g_{t_1}(f) = \frac{1}{t_1} \log \left( \int \exp(t_1 f(y)) \, dy \right) \tag{23}$$

and that softplus function is defined as

$$h_{t_2}(u) = \frac{1}{t_2} \log(1 + e^{t_2 u}). \tag{24}$$

We can now write the smoothed Makarov bounds as

$$\rho_{t_1,t_2}^-(\delta) = \mathbb{E}\left[\frac{1}{t_2} \log\left(1 + I_{t_1,\delta}(X)^{\frac{t_2}{t_1}}\right)\right] = \mathbb{E}\left[h_{t_2}\left(g_{t_1}(F_1(\cdot|X) - F_0(\cdot - \delta|X)))\right)\right] \tag{25}$$

and

$$\rho_{t_1,t_2}^+(\delta) = 1 - \mathbb{E}\left[\frac{1}{t_2} \log\left(1 + I_{-t_1,\delta}(X)^{\frac{t_2}{t_1}}\right)\right] = 1 + \mathbb{E}\left[h_{-t_2}\left(g_{-t_1}(F_1(\cdot|X) - F_0(\cdot - \delta|X)))\right)\right]. \tag{26}$$

We now employ standard bounds on the softmax and LSE functions. For all $u \in \mathbb{R}$, $t_2 > 0$, it holds that

$$h_{t_2}(u) - \frac{\log 2}{t_2} \leq (u)_+ \quad \text{and} \quad (u)_- \leq h_{-t_2}(u) + \frac{\log 2}{t_2}. \tag{27}$$

For bounded $f : \mathcal{Y} \to \mathbb{R}$ on compact $\mathcal{Y}$ and $t_1 > 0$, it holds that

$$g_{t_1}(f) - \frac{\log|\mathcal{Y}|}{t_1} \leq \sup_{y \in \mathcal{Y}} f(y) \quad \text{and} \quad \inf_{y \in \mathcal{Y}} f(y) \leq g_{-t_1}(f) + \frac{\log|\mathcal{Y}|}{t_1}. \tag{28}$$

In the following, we prove the theorem for the lower Makarov bound but the argument for the upper bound follows analogously. We apply the inequalities in (27) and (28) to Eq. (25) and obtain

$$\rho_{t_1,t_2}^-(\delta) \overset{(*)}{\leq} \mathbb{E}\left[h_{t_2}\left(\sup_{y \in \mathcal{Y}}(F_1(y|X) - F_0(y - \delta|X)) + \frac{\log|\mathcal{Y}|}{t_1}\right)\right] \tag{29}$$

$$\leq \mathbb{E}\left[\left(\sup_{y \in \mathcal{Y}}(F_1(y|X) - F_0(y - \delta|X)) + \frac{\log|\mathcal{Y}|}{t_1}\right)_+\right] + \frac{\log 2}{t_2} \tag{30}$$

$$\overset{(**)}{\leq} \rho^-(\delta) + \frac{\log|\mathcal{Y}|_+}{t_1} + \frac{\log 2}{t_2}, \tag{31}$$

where $(*)$ follows from monotonicity of $h_{t_2}$ and $(**)$ follows from $(a + b)_+ \leq a_+ + b_+$. $\qquad \square$

### A.2 PROOF OF THEOREM 4.2

*Proof.* We start by deriving the efficinet influence functions (EIFs) of the component $I_{t_1,\delta}(x)$. By employing the chain rule for influence functions (Kennedy et al., 2023), we obtain

$$EIF\{I_{t_1,\delta}(x)\} = t_1 \int_{\mathcal{Y}} \exp\left(t_1 \left[F_1(y \mid x) - F_0(y - \delta \mid x)\right]\right) EIF\{F_1(y \mid x) - F_0(y - \delta \mid x)\} \, dy. \tag{32}$$

Hence, it holds that

$$EIF\{I_{t_1,\delta}(x)^{\frac{t_2}{t_1}}\} = \frac{t_2}{t_1} I_{t_1,\delta}(x)^{\frac{t_2}{t_1} - 1} EIF\{I_{t_1,\delta}(x)\} \tag{33}$$

$$= t_2 I_{t_1,\delta}(x)^{\frac{t_2}{t_1}} \int_{\mathcal{Y}} w_{t_1,\delta}(y|x) EIF\{F_1(y \mid x) - F_0(y - \delta \mid x)\} \, dy, \tag{34}$$

and for the upper bound analogue

$$EIF\{I_{-t_1,\delta}(x)^{\frac{t_2}{t_1}}\} = \frac{t_2}{t_1}I_{-t_1,\delta}(x)^{\frac{t_2}{t_1}-1}\,EIF\{I_{-t_1,\delta}(x)\} \tag{35}$$

$$= -t_2 I_{-t_1,\delta}(x)^{\frac{t_2}{t_1}}\int_{\mathcal{Y}} w_{-t_1,\delta}(y|x)\,EIF\{F_1(y\mid x) - F_0(y-\delta\mid x)\}\,dy. \tag{36}$$

The efficient influence functions for the treatment and control c.d.f. are of standard form (Melnychuk et al., 2024), i.e.,

$$EIF\{F_a(y-\delta|X)\} = \frac{\mathbf{1}(A=a)\delta(X=x)}{\mathbb{P}(X=x,A=a)}\left(\mathbf{1}(Y\le y-\delta) - F_a(y-\delta|x)\right). \tag{37}$$

Plugging everything together, the EIF for the lower smoothed Makarov bound is

$$\Psi^{-}_{t_1,t_2,\delta}(Z,\eta) \tag{38}$$

$$= \int \frac{\mathbb{P}(X=x)}{t_2} EIF \log\left(1 + I_{t_1,\delta}(x)^{\frac{t_2}{t_1}}\right) dx \tag{39}$$

$$+ \frac{1}{t_2}\log\left(1 + I_{t_1,\delta}(X)^{\frac{t_2}{t_1}}\right) - \rho^{-}_{t_1,t_2}(\delta) \tag{40}$$

$$= \int \frac{\mathbb{P}(X=x)}{t_2}\frac{EIF\,I_{t_1,\delta}(x)^{\frac{t_2}{t_1}}}{1 + I_{t_1,\delta}(x)^{\frac{t_2}{t_1}}}\,dx + \frac{1}{t_2}\log\left(1 + I_{t_1,\delta}(X)^{\frac{t_2}{t_1}}\right) - \rho^{-}_{t_1,t_2}(\delta) \tag{41}$$

$$= \int \Pr(X=x)\underbrace{\frac{I_{t_1,\delta}(x)^{\frac{t_2}{t_1}}}{1 + I_{t_1,\delta}(x)^{\frac{t_2}{t_1}}}}_{\sigma^{-}_{t_1,t_2,\delta}(x)}\left[\int_{\mathcal{Y}} w_{t_1,\delta}(y|x)\,EIF\{F_1 - F_0\}\,dy\right]dx \tag{42}$$

$$+ \frac{1}{t_2}\log\left(1 + I_{t_1,\delta}(X)^{\frac{t_2}{t_1}}\right) - \rho^{-}_{t_1,t_2}(\delta) \tag{43}$$

$$= \int \Pr(X=x)\,\sigma^{-}_{t_1,t_2,\delta}(x)\left[\int_{\mathcal{Y}} w_{t_1,\delta}(y|x)\frac{A\delta(X=x)}{\mathbb{P}(X=x,A=1)}\left(\mathbf{1}(Y\le y) - F_1(y|x)\right)\,dy\right]dx \tag{44}$$

$$- \int \Pr(X=x)\,\sigma^{-}_{t_1,t_2,\delta}(x)\left[\int_{\mathcal{Y}} w_{t_1,\delta}(y|x)\frac{(1-A)\delta(X=x)}{\mathbb{P}(X=x,A=0)}\left(\mathbf{1}(Y\le y-\delta) - F_0(y-\delta|x)\right)\,dy\right]dx \tag{45}$$

$$+ \frac{1}{t_2}\log\left(1 + I_{t_1,\delta}(X)^{\frac{t_2}{t_1}}\right) - \rho^{-}_{t_1,t_2}(\delta) \tag{46}$$

$$= \sigma^{-}_{t_1,t_2,\delta}(X)\left[\int_{\mathcal{Y}} w_{t_1,\delta}(y|X)\frac{A}{\pi(X)}\left(\mathbf{1}(Y\le y) - F_1(y|X)\right)\,dy\right] \tag{47}$$

$$- \sigma^{-}_{t_1,t_2,\delta}(X)\left[\int_{\mathcal{Y}} w_{t_1}(y|X)\frac{(1-A)}{1-\pi(X)}\left(\mathbf{1}(Y\le y-\delta) - F_0(y-\delta|X)\right)\,dy\right] \tag{48}$$

$$+ \frac{1}{t_2}\log\left(1 + I_{t_1,\delta}(X)^{\frac{t_2}{t_1}}\right) - \rho^{-}_{t_1,t_2}(\delta) \tag{49}$$

$$\tag{50}$$

Analogously, we obtain the EIF for the upper smoothed Makarov bound as

$$\Psi_{t_1,t_2,\delta}^{+}(Z,\eta) \tag{51}$$

$$= \int \mathbb{P}(X=x) EIF \left\{ 1 - \frac{1}{t_2} \log\left(1 + I_{-t_1,\delta}(x)^{\frac{t_2}{t_1}}\right) \right\} dx \tag{52}$$

$$+ 1 - \frac{1}{t_2} \log\left(1 + I_{-t_1,\delta}(X)^{\frac{t_2}{t_1}}\right) - \rho_{t_1,t_2}^{+}(\delta) \tag{53}$$

$$= \int \frac{\mathbb{P}(X=x)}{-t_2} EIF \log\left(1 + I_{-t_1,\delta}(x)^{\frac{t_2}{t_1}}\right) dx \tag{54}$$

$$+ 1 - \frac{1}{t_2} \log\left(1 + I_{-t_1,\delta}(X)^{\frac{t_2}{t_1}}\right) - \rho_{t_1,t_2}^{+}(\delta) \tag{55}$$

$$= \int \frac{\mathbb{P}(X=x)}{-t_2} \frac{EIF I_{-t_1,\delta}(x)^{\frac{t_2}{t_1}}}{1 + I_{-t_1,\delta}(x)^{\frac{t_2}{t_1}}} dx + 1 - \frac{1}{t_2} \log\left(1 + I_{-t_1,\delta}(X)^{\frac{t_2}{t_1}}\right) - \rho_{t_1,t_2}^{+}(\delta) \tag{56}$$

$$= \int \Pr(X=x) \underbrace{\frac{I_{-t_1,\delta}(x)^{\frac{t_2}{t_1}}}{1 + I_{-t_1,\delta}(x)^{\frac{t_2}{t_1}}}}_{\sigma_{t_1,t_2,\delta}^{+}(x)} \left[\int_{\mathcal{Y}} w_{-t_1,\delta}(y|x)\, EIF\{F_1 - F_0\}\, dy\right] dx \tag{57}$$

$$+ 1 - \frac{1}{t_2} \log\left(1 + I_{-t_1,\delta}(X)^{\frac{t_2}{t_1}}\right) - \rho_{t_1,t_2}^{+}(\delta) \tag{58}$$

$$= \int \Pr(X=x)\, \sigma_{t_1,t_2,\delta}^{+}(x) \left[\int_{\mathcal{Y}} w_{-t_1,\delta}(y|x) \frac{A\delta(X=x)}{\mathbb{P}(X=x,A=1)} \left(\mathbf{1}(Y \leq y) - F_1(y|x)\right)\, dy\right] dx \tag{59}$$

$$- \int \Pr(X=x)\, \sigma_{t_1,t_2,\delta}^{+}(x) \left[\int_{\mathcal{Y}} w_{-t_1,\delta}(y|x) \frac{(1-A)\delta(X=x)}{\mathbb{P}(X=x,A=0)} \left(\mathbf{1}(Y \leq y-\delta) - F_0(y-\delta|x)\right)\, dy\right] dx \tag{60}$$

$$+ 1 - \frac{1}{t_2} \log\left(1 + I_{-t_1,\delta}(X)^{\frac{t_2}{t_1}}\right) - \rho_{t_1,t_2}^{+}(\delta) \tag{61}$$

$$= \sigma_{t_1,t_2,\delta}^{+}(X) \left[\int_{\mathcal{Y}} w_{-t_1,\delta}(y|X) \frac{A}{\pi(X)} \left(\mathbf{1}(Y \leq y) - F_1(y|X)\right)\, dy\right] \tag{62}$$

$$- \sigma_{t_1,t_2,\delta}^{+}(X) \left[\int_{\mathcal{Y}} w_{-t_1}(y|X) \frac{(1-A)}{1-\pi(X)} \left(\mathbf{1}(Y \leq y-\delta) - F_0(y-\delta|X)\right)\, dy\right] \tag{63}$$

$$+ 1 - \frac{1}{t_2} \log\left(1 + I_{-t_1,\delta}(X)^{\frac{t_2}{t_1}}\right) - \rho_{t_1,t_2}^{+}(\delta). \tag{64}$$

$$\square$$

## A.3 Proof of Corollary 4.3

**Assumption A.1** (Regularity and rate conditions for Corollary 4.3)**.** Fix $\delta \in \mathbb{R}$ and smoothing parameters $t_1, t_2 > 0$. Let $Z = (X, A, Y) \sim \mathbb{P}$ denote a generic observation and let $\eta = (F_0, F_1, \pi)$ denote the collection of nuisance functions.

1. **Causal assumptions.** Assumption 3.1 holds: consistency, overlap, and ignorability.

2. **Outcome support and boundedness.** The outcome support $\mathcal{Y}$ is compact with finite Lebesgue measure, as in Lemma 4.1. The efficient influence functions $\Psi_{t_1,t_2,\delta}^{\pm}(Z, \eta, \theta^{\pm}(\delta))$ in Theorem 4.2 are square-integrable with strictly positive and finite variance

$$0 < V^{\pm}(\delta) := Var\left(\Psi_{t_1,t_2,\delta}^{\pm}(Z, \eta, \theta^{\pm}(\delta))\right) < \infty.$$

3. **Cross-fitting.** The estimators $\hat{\eta} = (\hat{F}_0, \hat{F}_1, \hat{\pi})$ are obtained with sample splitting / $K$-fold cross-fitting: for each observation $i$, the corresponding nuisance estimates $\hat{\eta}^{(-k(i))}$ are trained on all folds except the one containing $i$.

4. **Rate conditions for nuisances.** For $a \in \{0, 1\}$,

$$\|\hat{F}_a - F_a\|_2 = o_{\mathbb{P}}(n^{-1/4}), \qquad \|\hat{\pi} - \pi\|_2 = o_{\mathbb{P}}(n^{-1/4}),$$

where $\| \cdot \|_2$ denotes the $L_2(\mathbb{P})$ norm.

*Proof of Corollary 4.3.* Fix $\delta \in \mathbb{R}$ and abbreviate

$$\theta^{\pm}(\delta) = \rho_{t_1,t_2}^{\pm}(\delta), \qquad \hat{\theta}^{\pm}(\delta) = \hat{\rho}_{t_1,t_2}^{\pm}(\delta),$$

for the smoothed lower and upper Makarov bounds and their debiased estimators in (19). For readability we suppress the dependence on $\delta$ where no confusion arises.

**Step 1: Oracle influence function representation.** By Theorem 4.2 and Assumption A.1(v), the smoothed functionals $\theta^{\pm}$ are pathwise differentiable at $\mathbb{P}$ with efficient influence functions $\Psi_{t_1,t_2,\delta}^{\pm}(Z, \eta, \theta^{\pm}(\delta))$. Consider the oracle estimator

$$\tilde{\theta}^{\pm}(\delta) := \frac{1}{n} \sum_{i=1}^{n} \left( \theta^{\pm}(\delta) + \Psi_{t_1,t_2,\delta}^{\pm}(Z_i, \eta, \theta^{\pm}(\delta)) \right),$$

which satisfies

$$\tilde{\theta}^{\pm}(\delta) - \theta^{\pm}(\delta) = \frac{1}{n} \sum_{i=1}^{n} \Psi_{t_1,t_2,\delta}^{\pm}(Z_i, \eta, \theta^{\pm}(\delta)).$$

By Assumption A.1(ii), the summands are i.i.d. with mean zero and variance $V^{\pm}(\delta) \in (0, \infty)$, so the Lindeberg–Feller central limit theorem yields

$$\sqrt{n}\big(\tilde{\theta}^{\pm}(\delta) - \theta^{\pm}(\delta)\big) \Rightarrow \mathcal{N}\big(0, V^{\pm}(\delta)\big). \tag{65}$$

**Step 2: Effect of plug-in nuisance estimation with cross-fitting.** Let the sample be partitioned into $K$ folds and denote by $\hat{\eta}^{(-k)}$ the nuisance estimates trained on all folds except fold $k$. For each $i$, let $k(i)$ be the index of the fold containing $Z_i$. The debiased estimators with cross-fitting can be written as

$$\hat{\theta}^{\pm}(\delta) = \frac{1}{n} \sum_{i=1}^{n} \left( \theta^{\pm}(\delta) + \Psi_{t_1,t_2,\delta}^{\pm}(Z_i, \hat{\eta}^{(-k(i))}, \theta^{\pm}(\delta)) \right) + R_n^{\pm},$$

where $R_n^{\pm}$ collects the higher-order terms arising from replacing $\theta^{\pm}(\delta)$ by $\hat{\theta}^{\pm}(\delta)$ inside the influence function.

By Neyman orthogonality of the score in Theorem 4.2 and Assumption A.1(iii)–(iv), standard arguments for orthogonal / debiased estimation (see, e.g., Chernozhukov et al. (2018)) imply that

$$\left| \frac{1}{n} \sum_{i=1}^{n} \left( \Psi_{t_1,t_2,\delta}^{\pm}(Z_i, \hat{\eta}^{(-k(i))}, \theta^{\pm}(\delta)) - \Psi_{t_1,t_2,\delta}^{\pm}(Z_i, \eta, \theta^{\pm}(\delta)) \right) \right| = o_{\mathbb{P}}(n^{-1/2}),$$

and that $R_n^{\pm} = o_{\mathbb{P}}(n^{-1/2})$ by a Taylor expansion in $\theta$. Hence we obtain the linear expansion

$$\hat{\theta}^{\pm}(\delta) - \theta^{\pm}(\delta) = \frac{1}{n} \sum_{i=1}^{n} \Psi_{t_1,t_2,\delta}^{\pm}(Z_i, \eta, \theta^{\pm}(\delta)) + r_n^{\pm}, \qquad r_n^{\pm} = o_{\mathbb{P}}(n^{-1/2}). \tag{66}$$

Comparing (66) with the oracle representation shows that the leading term is identical and the remainder is negligible at the $\sqrt{n}$ scale. Combining (65) and (66) and applying Slutsky's lemma gives, for each fixed $\delta$,

$$\sqrt{n}\big(\hat{\theta}^{\pm}(\delta) - \theta^{\pm}(\delta)\big) \Rightarrow \mathcal{N}\big(0, V^{\pm}(\delta)\big). \tag{67}$$

**Step 3: Consistency of the standard error estimator.** Define

$$\widehat{V}^{\pm}(\delta) = \frac{1}{n}\sum_{i=1}^{n}\left(\Psi^{\pm}_{t_1,t_2,\delta}(Z_i,\hat{\eta}^{(-k(i))},\hat{\theta}^{\pm}(\delta)) - \bar{\Psi}^{\pm}(\delta)\right)^2,$$

where

$$\bar{\Psi}^{\pm}(\delta) = \frac{1}{n}\sum_{i=1}^{n}\Psi^{\pm}_{t_1,t_2,\delta}(Z_i,\hat{\eta}^{(-k(i))},\hat{\theta}^{\pm}(\delta)),$$

and set $\widehat{\mathrm{se}}^{\pm}(\delta) = \sqrt{\widehat{V}^{\pm}(\delta)/n}$.

By the same orthogonality and rate conditions as in Step 2, the difference between

$$\Psi^{\pm}_{t_1,t_2,\delta}(Z_i,\hat{\eta}^{(-k(i))},\hat{\theta}^{\pm}(\delta)) \quad\text{and}\quad \Psi^{\pm}_{t_1,t_2,\delta}(Z_i,\eta,\theta^{\pm}(\delta))$$

is $o_{\mathbb{P}}(1)$ in $L_2$ uniformly in $i$. A law of large numbers under cross-fitting then yields

$$\widehat{V}^{\pm}(\delta) \xrightarrow{\mathbb{P}} V^{\pm}(\delta),$$

so $\widehat{\mathrm{se}}^{\pm}(\delta) \xrightarrow{\P} \sqrt{V^{\pm}(\delta)/n}$. Combining this with (67) and using Slutsky's lemma again, we obtain the studentized central limit theorem

$$\frac{\hat{\theta}^{\pm}(\delta) - \theta^{\pm}(\delta)}{\widehat{\mathrm{se}}^{\pm}(\delta)} \Rightarrow \mathcal{N}(0,1). \tag{68}$$

**Step 4: One-sided intervals for the smoothed bounds.** From (68), for each fixed $\delta$,

$$\Pr\left\{\theta^{-}(\delta) \geq \hat{\theta}^{-}(\delta) - z_{1-\alpha/2}\,\widehat{\mathrm{se}}^{-}(\delta)\right\} \to 1 - \alpha/2,$$

and

$$\Pr\left\{\theta^{+}(\delta) \leq \hat{\theta}^{+}(\delta) + z_{1-\alpha/2}\,\widehat{\mathrm{se}}^{+}(\delta)\right\} \to 1 - \alpha/2,$$

where $z_{1-\alpha/2}$ is the $(1-\alpha/2)$ quantile of the standard normal distribution.

**Step 5: Translating to the original Makarov bounds.** By Lemma 4.1, for all fixed $t_1,t_2 > 0$,

$$\theta^{-}(\delta) - b(t_1,t_2) \leq \rho(\delta) \leq \theta^{+}(\delta) + b(t_1,t_2),$$

where $b(t_1,t_2)$ is the approximation bias bound. On the intersection of the two one-sided events from Step 4, we have

$$\hat{\theta}^{-}(\delta) - z_{1-\alpha/2}\,\widehat{\mathrm{se}}^{-}(\delta) - b(t_1,t_2) \leq \rho(\delta) \leq \hat{\theta}^{+}(\delta) + z_{1-\alpha/2}\,\widehat{\mathrm{se}}^{+}(\delta) + b(t_1,t_2).$$

By the definition of $c^{-}_{t_1,t_2,\delta,\alpha}(\mathcal{D}_n)$ and $c^{+}_{t_1,t_2,\delta,\alpha}(\mathcal{D}_n)$, this event is precisely

$$c^{-}_{t_1,t_2,\delta,\alpha}(\mathcal{D}_n) \leq \rho(\delta) \leq c^{+}_{t_1,t_2,\delta,\alpha}(\mathcal{D}_n).$$

A Bonferroni argument then yields

$$\lim_{n\to\infty}\Pr\left\{c^{-}_{t_1,t_2,\delta,\alpha}(\mathcal{D}_n) \leq \rho(\delta) \leq c^{+}_{t_1,t_2,\delta,\alpha}(\mathcal{D}_n)\right\} \geq 1 - \alpha,$$

which proves the corollary. $\qquad\square$

## B EXTENDED RELATED WORK

**Semiparametric efficient causal inference.** Semiparametric efficiency theory (van der Vaart, 1998) and efficient-influence function-based estimators have a long tradition in causal inference (Kennedy, 2022). Examples include the AIPTW estimator (Robins et al., 1994), Targeted maximum-likelihood estimation (TMLE) (van der Laan & Rubin, 2006). These frameworks have been extended to various causal quantities and are the de-facto standard for modern causal effect estimation in many settings (van der Laan & Gruber, 2012; Chernozhukov et al., 2018; Foster & Syrgkanis, 2023; Kennedy, 2023).

**Bounds for partially identified causal quantities** In many situations, the causal parameter of interest is only *partially* identified. That is, we need to obtain bounds on the parameter of interest which we can then estimate with observational data Manski (1990). Several works have proposed methods for partial identification of causal quantities, including treatment effects in instrumental variable settings Balke & Pearl (1997); Kilbertus et al. (2020) and more general causal graphs Duarte et al. (2023); Balazadeh et al. (2022); Chen et al. (2023); Padh et al. (2023), and treatment effect risk Kallus (2023). A related stream of literature obtains bounds under so-called sensitivity models, which impose assumptions on the degree of non-identifiability Tan (2006); Jesson et al. (2021); Dorn & Guo (2022); Dorn et al. (2024); Yin et al. (2022); Frauen et al. (2023); Jin et al. (2023); Frauen et al. (2024).

## C  EXTENSION TO DISCRETE OUTCOMES.

Here we provide minimal changes needed to extend our methodology to discrete outcomes. Let $\mathcal{Y}_d = \{y_1 < \cdots < y_M\}$ denote a finite ordered support. For $a \in \{0, 1\}$, write $F_a(\cdot \mid x)$ for the right–continuous conditional CDF of $Y \mid (A = a, X = x)$ on $\mathcal{Y}_d$. All other notation is as in the main text.

**Discrete log-sum-exp and softplus.**  The softplus $h_{t_2}(u) = t_2^{-1} \log(1 + e^{t_2 u})$ is unchanged. We replace the continuous log-sum-exp with its discrete analogue

$$g_{t_1}^{(d)}(f) \; = \; \frac{1}{t_1} \log\Big( \sum_{y \in \mathcal{Y}_d} \exp\{t_1 f(y)\} \Big). \tag{69}$$

Accordingly, define the discrete normalizer

$$I_{t,\delta}^{(d)}(x) \; = \; \sum_{y \in \mathcal{Y}_d} \exp\Big( t\big[F_1(y \mid x) - F_0(y - \delta \mid x)\big] \Big). \tag{70}$$

**Smoothed Makarov bounds (discrete).**  Replacing the operators in Sec. 4.1 by (69)–(70) yields the discrete smoothed bounds

$$\rho_{t_1,t_2}^{-,(d)}(\delta) = \mathbb{E}\left[ \frac{1}{t_2} \log\Big( 1 + \big( I_{t_1,\delta}^{(d)}(X) \big)^{t_2/t_1} \Big) \right], \qquad \rho_{t_1,t_2}^{+,(d)}(\delta) = 1 - \mathbb{E}\left[ \frac{1}{t_2} \log\Big( 1 + \big( I_{-t_1,\delta}^{(d)}(X) \big)^{t_2/t_1} \Big) \right]. \tag{71}$$

The analogue of Lemma 4.1 holds with the discrete approximation bias

$$b_d(t_1, t_2) \; = \; \frac{\log 2}{t_2} + \frac{\log M}{t_1}, \tag{72}$$

i.e. $\rho_{t_1,t_2}^{-,(d)}(\delta) - b_d(t_1, t_2) \le \rho^-(\delta)$ and $\rho^+(\delta) \le \rho_{t_1,t_2}^{+,(d)}(\delta) + b_d(t_1, t_2)$, with $\rho_{t_1,t_2}^{\pm,(d)}(\delta) \to \rho^\pm(\delta)$ as $t_1, t_2 \to \infty$.

**Efficient influence functions (discrete).**  The EIFs in Theorem 4.2 carry over after replacing all integrals over $y$ by sums over $\mathcal{Y}_d$. Define the discrete softmax weights

$$w_{t_1,\delta}^{(d)}(y \mid x) \; = \; \frac{\exp\Big( t_1 \big[F_1(y \mid x) - F_0(y - \delta \mid x)\big] \Big)}{I_{t_1,\delta}^{(d)}(x)}, \qquad y \in \mathcal{Y}_d, \tag{73}$$

and the discrete analogue of $\Phi_{t_1,\delta}^a$,

$$\Phi_{t_1,\delta}^{a,(d)}(X, Y) \; = \; \sum_{y \in \mathcal{Y}_d} w_{t_1,\delta}^{(d)}(y \mid X) \Big( \mathbf{1}\{Y \le y - (1-a)\delta\} - F_a\big(y - (1-a)\delta \mid X\big) \Big). \tag{74}$$

With $\sigma_{t_1,t_2,\delta}^{\pm}(x)$ defined exactly as in the main text but using $I_{\pm t_1,\delta}^{(d)}(x)$, the EIFs are

$$\Psi_{t_1,t_2,\delta}^{-,(d)}\big(Z, \eta, \rho_{t_1,t_2}^{-,(d)}(\delta)\big) = \sigma_{t_1,t_2,\delta}^{-}(X) \left[ \frac{A}{\pi(X)} \Phi_{t_1,\delta}^{1,(d)}(X, Y) - \frac{1-A}{1-\pi(X)} \Phi_{t_1,\delta}^{0,(d)}(X, Y) \right]$$
$$+ \frac{1}{t_2} \log\Big( 1 + \big( I_{t_1,\delta}^{(d)}(X) \big)^{t_2/t_1} \Big) - \rho_{t_1,t_2}^{-,(d)}(\delta). \tag{75}$$

$$\Psi_{t_1,t_2,\delta}^{+,(d)}\big(Z, \eta, \rho_{t_1,t_2}^{+,(d)}(\delta)\big) = \sigma_{t_1,t_2,\delta}^{+}(X) \left[ \frac{A}{\pi(X)} \Phi_{-t_1,\delta}^{1,(d)}(X, Y) - \frac{1-A}{1-\pi(X)} \Phi_{-t_1,\delta}^{0,(d)}(X, Y) \right]$$
$$+ 1 - \frac{1}{t_2} \log\Big( 1 + \big( I_{-t_1,\delta}^{(d)}(X) \big)^{t_2/t_1} \Big) - \rho_{t_1,t_2}^{+,(d)}(\delta). \tag{76}$$

**Debiased estimators (discrete).**    The one-step estimators in Eq. (19) become

$$\hat{\rho}_{t_1,t_2}^{-,(d)}(\delta) = \frac{1}{n}\sum_{i=1}^{n}\left\{\frac{1}{t_2}\log\left(1 + \left(\hat{I}_{t_1,\delta}^{(d)}(x_i)\right)^{t_2/t_1}\right) + \hat{\sigma}_{t_1,t_2,\delta}^{-}(x_i)\,\frac{a_i - \hat{\pi}(x_i)}{\hat{\pi}(x_i)\{1 - \hat{\pi}(x_i)\}}\,\hat{\Phi}_{t_1,\delta}^{a_i,(d)}(x_i,y_i)\right\}.$$
(77)

$$\hat{\rho}_{t_1,t_2}^{+,(d)}(\delta) = \frac{1}{n}\sum_{i=1}^{n}\left\{1 - \frac{1}{t_2}\log\left(1 + \left(\hat{I}_{-t_1,\delta}^{(d)}(x_i)\right)^{t_2/t_1}\right) + \hat{\sigma}_{t_1,t_2,\delta}^{+}(x_i)\,\frac{a_i - \hat{\pi}(x_i)}{\hat{\pi}(x_i)\{1 - \hat{\pi}(x_i)\}}\,\hat{\Phi}_{-t_1,\delta}^{a_i,(d)}(x_i,y_i)\right\}.$$
(78)

Here $\hat{I}_{\pm t_1,\delta}^{(d)}$, $\hat{w}_{t_1,\delta}^{(d)}$, and $\hat{\Phi}_{\pm t_1,\delta}^{a_i,(d)}$ replace (70), (73), and (74) with estimated nuisances. Cross-fitting and variance estimation via the sample variance of the estimated EIF proceed unchanged.

# D    EXTENSION TO UNIFORMLY VALID CONFIDENCE INTERVALS

In this section, we extend Corollary 4.3 to construct confidence intervals for the treatment effect distribution that are *simultaneously* valid for all values of $\delta$ in a compact set $\Delta \subset \mathbb{R}$. The construction proceeds in two steps. First, we establish a functional central limit theorem (CLT) for the smoothed debiased estimators $\hat{\rho}^{\pm}_{t_1,t_2}(\delta)$ viewed as stochastic processes in $\delta$. Second, we use a multiplier bootstrap to approximate the distribution of the supremum of the corresponding Gaussian limit and combine this with the approximation bias bound $b(t_1, t_2)$ from Lemma 4.1 to obtain uniform confidence bands for the original Makarov bounds.

## D.1    UNIFORM ASYMPTOTIC LINEARITY AND FUNCTIONAL CLT

We focus on a compact interval $\Delta \subset \mathbb{R}$ of values of $\delta$ that are of substantive interest. Throughout this section, we treat the smoothing parameters $(t_1, t_2)$ as fixed and suppress their dependence where it is notationally convenient. We write

$$\mathbb{G}_n f \;=\; \frac{1}{\sqrt{n}} \sum_{i=1}^{n} \big( f(Z_i) - \mathbb{E}[f(Z)] \big)$$

for the empirical process indexed by a function $f$, and consider the class of efficient influence functions

$$\mathcal{F}^{\pm} \;=\; \big\{ \Psi^{\pm}_{t_1,t_2,\delta}(\,\cdot\,, \eta, \rho^{\pm}_{t_1,t_2}(\delta)) : \delta \in \Delta \big\}$$

as given in Theorem 4.2.

We impose the following regularity condition, which is a uniform version of the high-level conditions used in Corollary 4.3.

**Assumption D.1** (Uniform regularity of the EIF process). Let $\Delta \subset \mathbb{R}$ be compact and let $\Psi^{\pm}_{t_1,t_2,\delta}(Z, \eta, \rho^{\pm}_{t_1,t_2}(\delta))$ be as in Theorem 4.2. Assume:

1. **Uniform bounded second moments:**

$$\sup_{\delta \in \Delta} \mathbb{E}\big[ \Psi^{\pm}_{t_1,t_2,\delta}(Z, \eta, \rho^{\pm}_{t_1,t_2}(\delta))^2 \big] < \infty.$$

2. **Stochastic equicontinuity in $\delta$:** there exists a semi-metric $d$ on $\Delta$ such that, for any $\varepsilon > 0$,

$$\lim_{\gamma \to 0} \limsup_{n \to \infty} \mathbb{P}\Big( \sup_{d(\delta,\delta') \leq \gamma} \big| \mathbb{G}_n\big( \Psi^{\pm}_{t_1,t_2,\delta} - \Psi^{\pm}_{t_1,t_2,\delta'} \big) \big| > \varepsilon \Big) = 0.$$

3. **Donsker-type condition:** the class $\mathcal{F}^{\pm}$ is $P$-Donsker (or, more generally, satisfies an entropy condition that guarantees a functional CLT for $\mathbb{G}_n$ indexed by $\mathcal{F}^{\pm}$).

4. **Uniform orthogonality and nuisance rates:** let $\hat{\eta}$ be obtained via sample splitting as in Corollary 4.3. The asymptotic linear expansion

$$\hat{\rho}^{\pm}_{t_1,t_2}(\delta) = \rho^{\pm}_{t_1,t_2}(\delta) + \frac{1}{n} \sum_{i=1}^{n} \Psi^{\pm}_{t_1,t_2,\delta}(Z_i, \eta, \rho^{\pm}_{t_1,t_2}(\delta)) + r^{\pm}_n(\delta)$$

holds with a remainder satisfying

$$\sup_{\delta \in \Delta} \sqrt{n} \, \big| r^{\pm}_n(\delta) \big| \; \xrightarrow{P} \; 0.$$

Under Assumption D.1, we obtain a functional CLT for the smoothed lower and upper bounds following standard arguments in empirical process theory (Van Der Vaart & Wellner, 1996).

**Theorem D.2** (Functional CLT for smoothed Makarov bounds). *Let Assumption D.1 hold and let $\Delta \subset \mathbb{R}$ be compact. Then, in the space $\ell^{\infty}(\Delta)$ of bounded real-valued functions on $\Delta$,*

$$\big\{ \sqrt{n}\big( \hat{\rho}^{-}_{t_1,t_2}(\delta) - \rho^{-}_{t_1,t_2}(\delta) \big) \big\}_{\delta \in \Delta} \rightsquigarrow \{ G^{-}(\delta) \}_{\delta \in \Delta}, \tag{79}$$

$$\big\{ \sqrt{n}\big( \hat{\rho}^{+}_{t_1,t_2}(\delta) - \rho^{+}_{t_1,t_2}(\delta) \big) \big\}_{\delta \in \Delta} \rightsquigarrow \{ G^{+}(\delta) \}_{\delta \in \Delta}, \tag{80}$$

*where $G^\pm$ are mean-zero tight Gaussian processes with covariance functions*

$$\text{Cov}\big(G^\pm(\delta_1), G^\pm(\delta_2)\big) = \mathbb{E}\big[\Psi^\pm_{t_1,t_2,\delta_1}(Z, \eta, \rho^\pm_{t_1,t_2}(\delta_1))\, \Psi^\pm_{t_1,t_2,\delta_2}(Z, \eta, \rho^\pm_{t_1,t_2}(\delta_2))\big]$$

*for all $\delta_1, \delta_2 \in \Delta$.*

As a consequence of Theorem D.2, the random variables

$$\sup_{\delta \in \Delta} \big|\sqrt{n}\big(\hat{\rho}^\pm_{t_1,t_2}(\delta) - \rho^\pm_{t_1,t_2}(\delta)\big)\big|$$

converge in distribution to $\sup_{\delta \in \Delta} |G^\pm(\delta)|$. We now describe how to approximate the distribution of these suprema via a multiplier bootstrap.

### D.2  Multiplier bootstrap for uniform bands of smoothed bounds

Define the estimated influence function contributions

$$\hat{\psi}^\pm_i(\delta) = \Psi^\pm_{t_1,t_2,\delta}\big(Z_i, \hat{\eta}, \hat{\rho}^\pm_{t_1,t_2}(\delta)\big), \qquad i = 1, \ldots, n, \ \ \delta \in \Delta,$$

where $\hat{\eta}$ and $\hat{\rho}^\pm_{t_1,t_2}(\delta)$ are obtained as in Eq. (19). Let

$$\bar{\hat{\psi}}^\pm_n(\delta) = \frac{1}{n} \sum_{i=1}^n \hat{\psi}^\pm_i(\delta)$$

denote the empirical mean of these contributions.

Let $\xi_1, \ldots, \xi_n$ be i.i.d. multiplier weights with $\mathbb{E}[\xi_i] = 0$ and $\mathbb{E}[\xi_i^2] = 1$ (e.g., Rademacher or standard normal). The multiplier bootstrap process is defined by

$$\hat{G}^{\pm,*}_n(\delta) = \frac{1}{\sqrt{n}} \sum_{i=1}^n \xi_i\big(\hat{\psi}^\pm_i(\delta) - \bar{\hat{\psi}}^\pm_n(\delta)\big), \qquad \delta \in \Delta. \tag{81}$$

We then consider the sup-norm statistic

$$T^{\pm,*}_n = \sup_{\delta \in \Delta} \big|\hat{G}^{\pm,*}_n(\delta)\big|.$$

Repeating this construction $B$ times with independent multipliers $\{\xi_i^{(b)}\}_{i=1}^n$, $b = 1, \ldots, B$, yields bootstrap draws

$$T^{\pm,*}_{n,1}, \ldots, T^{\pm,*}_{n,B}.$$

Let $\hat{c}^\pm_{1-\alpha}$ denote the empirical $(1-\alpha)$ quantile of $\{T^{\pm,*}_{n,b}\}_{b=1}^B$. Under standard conditions for the validity of the multiplier bootstrap for suprema of empirical processes (which are implied by Assumption D.1 and mild additional technical assumptions), this quantity consistently estimates the $(1-\alpha)$ quantile of $\sup_{\delta \in \Delta} |G^\pm(\delta)|$.

The following corollary summarizes the resulting uniform confidence bands for the *smoothed* bounds $\rho^\pm_{t_1,t_2}(\delta)$, following standard multiplier bootstrap theory for empirical processes (Van Der Vaart & Wellner, 1996).

**Corollary D.3** (Simultaneous confidence bands for smoothed Makarov bounds)**.** *Suppose the conditions of Theorem D.2 hold and the multiplier bootstrap described in Eq. (81) is valid for the processes in (79)–(80). Let $\hat{c}^\pm_{1-\alpha}$ be the empirical $(1-\alpha)$ quantiles of $T^{\pm,*}_n$ as defined above. Then,*

$$\lim_{n \to \infty} \mathbb{P}\left(\forall \delta \in \Delta : \ \big|\hat{\rho}^\pm_{t_1,t_2}(\delta) - \rho^\pm_{t_1,t_2}(\delta)\big| \leq \frac{\hat{c}^\pm_{1-\alpha}}{\sqrt{n}}\right) \geq 1 - \alpha.$$

*Equivalently, the bands*

$$\delta \mapsto \left[\hat{\rho}^\pm_{t_1,t_2}(\delta) - \frac{\hat{c}^\pm_{1-\alpha}}{\sqrt{n}}, \ \hat{\rho}^\pm_{t_1,t_2}(\delta) + \frac{\hat{c}^\pm_{1-\alpha}}{\sqrt{n}}\right]$$

*are asymptotically valid $(1-\alpha)$ simultaneous confidence bands for the smoothed Makarov bounds $\rho^\pm_{t_1,t_2}(\delta)$, uniformly over $\delta \in \Delta$.*

## D.3 Uniformly valid bands for the original Makarov bounds

We now translate the simultaneous bands for the smoothed bounds $\rho_{t_1,t_2}^{\pm}(\delta)$ into simultaneous bands for the original Makarov bounds $\rho^{\pm}(\delta)$. Recall from Lemma 4.1 that

$$\rho_{t_1,t_2}^{-}(\delta) - b(t_1, t_2) \leq \rho^{-}(\delta) \quad \text{and} \quad \rho^{+}(\delta) \leq \rho_{t_1,t_2}^{+}(\delta) + b(t_1, t_2), \tag{82}$$

for every $\delta$, where

$$b(t_1, t_2) \;=\; \frac{\log(2)}{t_2} + \frac{(\log|\mathcal{Y}|)_+}{t_1}$$

does not depend on $\delta$. Hence, the inequalities in (82) hold uniformly for all $\delta \in \Delta$.

Combining Corollary D.3 with (82) yields the following result.

**Corollary D.4** (Simultaneous confidence bands for the treatment effect distribution)**.** *Fix $\alpha \in (0, 1)$ and a compact set $\Delta \subset \mathbb{R}$. Let $\hat{c}_{1-\alpha/2}^{-,\mathrm{unif}}$ and $\hat{c}_{1-\alpha/2}^{+,\mathrm{unif}}$ denote the bootstrap critical values obtained as in Corollary D.3 for the lower and upper smoothed bounds, respectively, with level $\alpha/2$. Define, for each $\delta \in \Delta$,*

$$\underline{c}_n(\delta) = \hat{\rho}_{t_1,t_2}^{-}(\delta) - \frac{\hat{c}_{1-\alpha/2}^{-,\mathrm{unif}}}{\sqrt{n}} - b(t_1, t_2), \tag{83}$$

$$\bar{c}_n(\delta) = \hat{\rho}_{t_1,t_2}^{+}(\delta) + \frac{\hat{c}_{1-\alpha/2}^{+,\mathrm{unif}}}{\sqrt{n}} + b(t_1, t_2). \tag{84}$$

*Then*

$$\liminf_{n\to\infty} \mathbb{P}(\forall \delta \in \Delta : \; \underline{c}_n(\delta) \leq \rho(\delta) \leq \bar{c}_n(\delta)) \;\geq\; 1 - \alpha.$$

*In particular, the pair of bands $\{\underline{c}_n(\delta)\}_{\delta\in\Delta}$ and $\{\bar{c}_n(\delta)\}_{\delta\in\Delta}$ defines an asymptotically valid $(1 - \alpha)$ simultaneous confidence band for the treatment effect distribution $\rho(\delta)$ uniformly over $\delta \in \Delta$.*

**Practical implementation.** In practice, the set $\Delta$ is approximated by a finite grid $\{\delta_1, \ldots, \delta_K\} \subset \Delta$. The procedure then reduces to:

1. Compute $\hat{\rho}_{t_1,t_2}^{\pm}(\delta_k)$ and $\hat{\psi}_i^{\pm}(\delta_k)$ for all $k = 1, \ldots, K$.

2. For each bootstrap replication $b = 1, \ldots, B$, draw multipliers $\{\xi_i^{(b)}\}_{i=1}^n$ and form

$$\hat{G}_{n,b}^{\pm,*}(\delta_k) = \frac{1}{\sqrt{n}} \sum_{i=1}^{n} \xi_i^{(b)} \big(\hat{\psi}_i^{\pm}(\delta_k) - \bar{\hat{\psi}}_n^{\pm}(\delta_k)\big),$$

   and set $T_{n,b}^{\pm,*} = \max_{k=1,\ldots,K} |\hat{G}_{n,b}^{\pm,*}(\delta_k)|$.

3. Let $\hat{c}_{1-\alpha/2}^{\pm,\mathrm{unif}}$ be the empirical $(1 - \alpha/2)$ quantiles of $\{T_{n,b}^{\pm,*}\}_{b=1}^B$, and construct $\underline{c}_n(\delta_k)$ and $\bar{c}_n(\delta_k)$ as in (83)–(84).

For a sufficiently dense grid, the resulting discrete bands provide an accurate approximation to the uniform confidence bands over $\Delta$.

# E    EXTENSION TO MULTIVALUED TREATMENTS

In this appendix, we briefly describe how our methodology extends to settings with multivalued treatments. Let the treatment take values in a finite set $\mathcal{A} \subset \mathbb{R}$ with $|\mathcal{A}| \geq 2$, and let

$$Z = (X, A, Y) \sim \mathbb{P}, \qquad A \in \mathcal{A},$$

where $X \in \mathcal{X} \subseteq \mathbb{R}^d$ and $Y \in \mathbb{R}$ as in the main text. For each $a \in \mathcal{A}$ we denote the potential outcome by $Y(a)$ and define the generalized propensity scores and response distributions by

$$\pi_a(x) = \mathbb{P}(A = a \mid X = x), \qquad F_a(y \mid x) = \mathbb{P}(Y \leq y \mid X = x, A = a).$$

## E.1    TARGET ESTIMAND FOR A PAIR OF TREATMENT LEVELS

In many applications, the primary goal is to compare two specific treatment levels $a_1, a_0 \in \mathcal{A}$. For any such ordered pair $(a_1, a_0)$ we define the treatment effect distribution

$$\rho_{a_1, a_0}(\delta) = \mathbb{P}\big(Y(a_1) - Y(a_0) \leq \delta\big), \qquad \delta \in \mathbb{R}. \tag{85}$$

This is directly analogous to the binary-treatment estimand in the main text, where $(a_1, a_0) = (1, 0)$.

Under the natural multivalued analogue of Assumption 3.1, namely:

1. *Consistency:* $Y(a) = Y$ whenever $A = a$ for all $a \in \mathcal{A}$,

2. *Overlap:* $0 < \pi_a(X) < 1$ almost surely for all $a \in \mathcal{A}$,

3. *Ignorability:* $A \perp \{Y(a) : a \in \mathcal{A}\} \mid X$,

the treatment effect distribution $\rho_{a_1, a_0}(\delta)$ is again partially identified by Makarov-type bounds based on the pair of marginals $(F_{a_1}, F_{a_0})$:

$$\rho_{a_1, a_0}^-(\delta) \leq \rho_{a_1, a_0}(\delta) \leq \rho_{a_1, a_0}^+(\delta), \tag{86}$$

where, for each fixed pair $(a_1, a_0)$,

$$\rho_{a_1, a_0}^-(\delta) = \mathbb{E}\left[\sup_{y \in \mathcal{Y}} \big(F_{a_1}(y \mid X) - F_{a_0}(y - \delta \mid X)\big)_+\right], \tag{87}$$

and

$$\rho_{a_1, a_0}^+(\delta) = 1 + \mathbb{E}\left[\inf_{y \in \mathcal{Y}} \big(F_{a_1}(y \mid X) - F_{a_0}(y - \delta \mid X)\big)_-\right], \tag{88}$$

which are obtained by replacing $F_1$ and $F_0$ in Eq. (3) with $F_{a_1}$ and $F_{a_0}$.

## E.2    REDUCTION TO THE BINARY CASE

Our proposed estimators and theoretical results for the binary-treatment case extend directly to this multivalued setting by a simple recoding argument. For a fixed pair $(a_1, a_0)$, define the binary indicators

$$\tilde{A}_{a_1, a_0} = \mathbf{1}\{A = a_1\}, \qquad 1 - \tilde{A}_{a_1, a_0} = \mathbf{1}\{A = a_0\},$$

and the corresponding generalized propensity scores

$$\pi_{a_1}(X) = \mathbb{P}(A = a_1 \mid X), \qquad \pi_{a_0}(X) = \mathbb{P}(A = a_0 \mid X).$$

Conditional on $X$, the distribution of $(\tilde{A}_{a_1, a_0}, Y)$ restricted to the subset $\{A \in \{a_1, a_0\}\}$ is algebraically identical to the binary-treatment setup in the main text with treatment $A = 1$ and control $A = 0$, after replacing:

$$A \rightsquigarrow \mathbf{1}\{A = a_1\}, \quad 1 - A \rightsquigarrow \mathbf{1}\{A = a_0\}, \quad \pi(X) \rightsquigarrow \pi_{a_1}(X), \quad 1 - \pi(X) \rightsquigarrow \pi_{a_0}(X),$$

and

$$F_1(\cdot \mid X) \rightsquigarrow F_{a_1}(\cdot \mid X), \qquad F_0(\cdot \mid X) \rightsquigarrow F_{a_0}(\cdot \mid X).$$

### E.3 INFLUENCE FUNCTIONS AND DEBIASED ESTIMATORS

Let $\rho^{\pm}_{a_1,a_0,t_1,t_2}(\delta)$ denote the smoothed Makarov bounds obtained by applying the smoothing construction from Lemma 4.1 to $(F_{a_1}, F_{a_0})$ instead of $(F_1, F_0)$. The corresponding efficient influence functions follow by the same substitutions in Theorem 4.2.

*Remark* E.1 (EIF for multivalued treatments). Fix $a_1, a_0 \in \mathcal{A}$ and define

$$d_{a_1,a_0,\delta,\eta}(y \mid X) \;=\; F_{a_1}(y \mid X) - F_{a_0}(y - \delta \mid X),$$

and $I^{(a_1,a_0)}_{t,\delta}(x)$, $w^{(a_1,a_0)}_{t_1,\delta}(y \mid x)$, $\sigma^{\pm,(a_1,a_0)}_{t_1,t_2,\delta}(x)$ analogously to $I_{t,\delta}(x)$, $w_{t_1,\delta}(y \mid x)$ and $\sigma^{\pm}_{t_1,t_2,\delta}(x)$ in Theorem 4.2, but with $F_1, F_0$ replaced by $F_{a_1}, F_{a_0}$. Then the efficient influence functions for the smoothed multivalued bounds $\rho^{\pm}_{a_1,a_0,t_1,t_2}(\delta)$ are obtained from Theorem 4.2 by replacing

$$A \rightarrow \mathbf{1}\{A = a_1\}, \;\; 1 - A \rightarrow \mathbf{1}\{A = a_0\}, \;\; \pi(X) \rightarrow \pi_{a_1}(X), \;\; 1 - \pi(X) \rightarrow \pi_{a_0}(X), \;\; F_1 \rightarrow F_{a_1}, \;\; F_0 \rightarrow F_{a_0}.$$

In particular, the debiased estimators in Eq. (19) extend verbatim to the comparison of any pair $(a_1, a_0)$ after making these substitutions.

Because the derivation of the EIF in Theorem 4.2 relies only on the binary nature of the comparison (treated vs. control) and not on the cardinality of $\mathcal{A}$, the same arguments imply that all results on asymptotic normality, confidence intervals, and uniform confidence bands (Corollary 4.3 and Appendix D) carry over to the multivalued setting for any fixed pair $(a_1, a_0)$.

In practice, a practitioner who wishes to compare the treatment levels $a_1$ and $a_0$ simply specifies these two values and applies our binary-treatment procedure to the recoded data with $A$ replaced by $\mathbf{1}\{A = a_1\}$ and the nuisance functions $(F_1, F_0, \pi)$ replaced by $(F_{a_1}, F_{a_0}, \pi_{a_1})$ and $\pi_{a_0}$, as described above.

# F  DETAILS ON NUISANCE ESTIMATION

We estimate the response c.d.f.s $F_a(y \mid x)$, $a \in \{0, 1\}$, by fitting separate conditional distribution models within each arm ($A = a$). All learners return the *entire* conditional c.d.f. evaluated on an arbitrary grid $y \in \mathcal{Y}$, which is required by the EIF in (4.2).

**Training protocol (common to all learners).**  For each arm $a$ we split the data into two folds as outlined in Algorithm 1. On each fold, we train on the complement and predict $\hat{F}_a(\cdot \mid x_i)$ for held-out $i$. Models are trained by gradient boosting (LightGBM) with early stopping on a validation set and a likelihood-based metric. Monotone transformations $T$ of $Y$ can be applied for numerical stability; since $T$ is monotone, $F_Y(y \mid x) = F_{T(Y)}(T(y) \mid x)$. In particular, we use standardization and log-transformation where appropriate. Given a grid $\{y_j\}_{j=1}^{J}$, each learner returns the matrix $\left(\hat{F}_a(y_j \mid x_i)\right)_{i,j}$. These values feed the smoothed operators in (9) and the weighted integrals $\hat{\Phi}_{t_1, \delta}^{a}(x_i, y_i)$ in (19).

**Continuous outcomes: conditional Gaussian mixtures.**  For $Y \in \mathbb{R}$ we model $T(Y) \mid X = x$ as a $K$-component Gaussian mixture with covariate-dependent weights $\pi_k(x)$, means $\mu_k(x)$ and variances $\sigma_k^2(x)$. Boosting optimizes the (negative) log-likelihood with a custom objective

$$\ell_{\mathrm{GM}}(x, y) = -\log\left(\sum_{k=1}^{K} \pi_k(x)\, \varphi\big(T(y); \mu_k(x), \sigma_k^2(x)\big)\right),$$

where $\varphi(y; \mu, \sigma^2)$ is the $\mathcal{N}(\mu, \sigma^2)$ density. The log-likelihood loss supplies per-parameter gradients and diagonal Hessians. Derivatives are rescaled to balance curvature across logits/means/variances. Initialization uses $k$-means on $T(Y)$ (cluster means/variances feed the intial score matrix. The resulting c.d.f. is

$$\hat{F}_a(y \mid x) = \sum_{k=1}^{K} \hat{\pi}_k(x)\, \Phi\left(\frac{T(y) - \hat{\mu}_k(x)}{\hat{\sigma}_k(x)}\right).$$

**Discrete outcomes: multinomial classifier.**  For discrete $Y$ taking finitely many values $\{c_1 < \cdots < c_M\}$ we fit a multiclass boosted classifier returning $\hat{p}_x(c_m) = \Pr(Y = c_m \mid X = x)$. The c.d.f. is the cumulative sum

$$\hat{F}_a(y \mid x) = \sum_{m:\, c_m \leq y} \hat{p}_x(c_m),$$

implemented by summing predicted class probabilities over classes $\leq y$.

**Counts with excess zeros: zero–inflated Poisson (ZIP).**  For nonnegative counts we fit a covariate-dependent ZIP with rate $\lambda(x)$ and zero–inflation $\psi(x)$, learned via a custom objective for the ZIP log-likelihood and stabilized derivatives. Initialization uses the empirical mean for $\lambda$ and the observed zero rate for $\psi$. The c.d.f. (right-continuous) is

$$\hat{F}_a(y \mid x) = \begin{cases} 0, & y < 0, \\ \psi(x) + (1 - \psi(x))\, \mathrm{PoisCDF}(\lfloor y \rfloor; \lambda(x)), & y \geq 0. \end{cases}$$

## G  Details Regarding Synthetic Data

**Data generating process.**

We sample uniform covariates $X \sim \mathrm{Unif}[0,1]$ and define $\rho(x) = \frac{1}{2}x^2 + \frac{1}{2}$. We then define the c.d.f. for the treatment arm as

$$Y(1) \mid X = x \sim \mathrm{Unif}\big([0, \rho(x)]\big), \qquad F_{1|x}(y) = \begin{cases} 0, & y < 0, \\ \dfrac{y}{\rho(x)}, & 0 \le y \le \rho(x), \\ 1, & y > \rho(x). \end{cases}$$

For the control arm, we define for a parameter $\gamma$

$$L_1(x) = -\tfrac{\gamma}{4}\,x, \qquad L_2(x) = \tfrac{1}{2}\rho(x) + \tfrac{\gamma}{4}\,x.$$

Then, we define the control arm c.d.f. as

$$F_{0|x}(y) = \begin{cases} 0, & y < L_1, \\ \dfrac{y - L_1}{\rho}, & L_1 \le y \le L_1 + \tfrac{1}{2}\rho, \\ \tfrac{1}{2}, & L_1 + \tfrac{1}{2}\rho < y < L_2, \\ \tfrac{1}{2} + \dfrac{y - L_2}{\rho}, & L_2 \le y \le L_2 + \tfrac{1}{2}\rho, \\ 1, & y > L_2 + \tfrac{1}{2}\rho, \end{cases}$$

where, for brevity, $L_j = L_j(x)$ and $\rho = \rho(x)$. The gap between the two control components is

$$G(x) = L_2 - \left(L_1 + \tfrac{1}{2}\rho\right) = \frac{\gamma}{2}\,x.$$

To model a joint distribution leveraging the marginals above, we draw $(U_1, U_0)$ from a Gumbel copula $C_\theta$ with $\theta = 5$ (dimension 2), and set

$$Y(1) \mid X = F_{1|X}^{-1}(U_1), \qquad Y(0) \mid X = F_{0|X}^{-1}(U_0).$$

Finally, the observed outcome is $Y = A\,Y(1) + (1 - A)\,Y(0)$.

**Quantifying margin violation.** The difference $D_x(y) = F_{1|x}(y) - F_{0|x}(y)$ admits two plateaus that attain supremum and infimum and thus violating the margin assumption. The left, negative one is given for $0 \le y \le \frac{1}{2}\rho - \frac{\gamma}{4}x$ as

$$D_x(y) = \frac{y}{\rho} - \frac{y - L_1}{\rho} = \frac{L_1}{\rho} = -\frac{\gamma x}{4\,\rho}.$$

Its width is

$$W_{\mathrm{L}}(x) = \frac{\rho}{2} - \frac{\gamma}{4}x.$$

The right, positive plateau is given for $\frac{1}{2}\rho + \frac{\gamma}{4}x \le y \le \rho$ as

$$D_x(y) = \frac{y}{\rho} - \left(\tfrac{1}{2} + \frac{y - L_2}{\rho}\right) = \frac{L_2 - \tfrac{1}{2}\rho}{\rho} = +\frac{\gamma x}{4\,\rho}.$$

Its width is given by

$$W_{\mathrm{R}}(x) = \frac{\rho}{2} - \frac{\gamma}{4}x.$$

Averaging over $X$ yields $\mathbb{E}[\rho(X)] = \frac{2}{3}$, $\mathbb{E}[X] = \frac{1}{2}$, and

$$\mathbb{E}\left[\frac{X}{\rho(X)}\right] = \int_0^1 \frac{2x}{x^2 + 1}\, dx = \ln 2.$$

Hence, obtain average plateau widths

$$\mathbb{E}\big[W_{\mathrm{L}}(X)\big] = \mathbb{E}\big[W_{\mathrm{R}}(X)\big] = \frac{1}{3} - \frac{\gamma}{8}, \qquad \mathbb{E}\big[G(X)\big] = \frac{\gamma}{4},$$

and normalized

$$\mathbb{E}\left[\frac{W_{\mathrm{L}}(X)}{\rho(X)}\right] = \mathbb{E}\left[\frac{W_{\mathrm{R}}(X)}{\rho(X)}\right] = \frac{1}{2} - \frac{\gamma}{4}\ln 2, \qquad \mathbb{E}\left[\frac{G(X)}{\rho(X)}\right] = \frac{\gamma}{2}\ln 2.$$

Thus, when descreasing $\gamma$, we maximize the average platenau width and correspondingly the degree of margin violation.

## H    DETAILS REGARDING SEMI-SYNTHETIC DATA

**Real-world covariate data.** The so-called *Oregon health insurance experiment*[2] (OHIE) (Finkelstein et al., 2012) is a randomized experiment that was intentionally conducted as a large-scale effort in public health to assess the effect of health insurance on several outcomes such as health or economic status. In 2008, a lottery draw offered low-income, uninsured adults in Oregon participation in a Medicaid program, providing health insurance. Individuals whose names were drawn could decide to sign up for the program.

In our analysis, we extract the following covariates $X$: age, gender, language, the number of emergency visits before the experiment, and the number of people the individual signed up with. The data collection in the OHIE was done as follows: after excluding individuals below the age of 19, above the age of 64, and individuals with residence outside of Oregon, 74,922 individuals were considered for the lottery. Among those, 29,834 were selected randomly and were offered participation in the program. Out of these, we randomly select $n = 3000$ data points.

**Synthetic treatment and outcome generation.** We define the empirical mean covariate

$$m(x) \;=\; \frac{1}{p} \sum_{j=1}^{p} x_j.$$

and introduce two covariate-dependent shape functions:

$$\rho(x) \;=\; \tfrac{1}{2}\, m(x)^2 + \rho_0, \qquad g(x) \;=\; m(x)^3,$$

where $\rho_0 > 0$ is a constant hyperparameter (in our experiments $\rho_0 = 0.5$). The treatment is randomized with constant propensity:

$$\pi(x) \equiv \mathbb{P}(A = 1 \mid X = x) = \tfrac{1}{2}, \qquad A_i \sim \text{Bernoulli}\!\left(\tfrac{1}{2}\right) \text{ independently of } X_i.$$

We define the treatment c.d.f. as

$$Y(1) \mid X = x \sim \text{Unif}\!\left([0,\, \rho(x)]\right), \qquad F_{1|x}(y) = \begin{cases} 0, & y < 0, \\[4pt] \dfrac{y}{\rho(x)}, & 0 \le y \le \rho(x), \\[4pt] 1, & y > \rho(x). \end{cases}$$

For the control arm c.d.f., we define

$$L_1(x) = -\tfrac{1}{4}\, g(x), \qquad L_2(x) = \tfrac{1}{2}\rho(x) + \tfrac{1}{4}\, g(x),$$

and

$$F_{0|x}(y) = \begin{cases} 0, & y < L_1, \\[4pt] \dfrac{y - L_1}{\rho}, & L_1 \le y \le L_1 + \tfrac{1}{2}\rho, \\[4pt] \tfrac{1}{2}, & L_1 + \tfrac{1}{2}\rho < y < L_2, \\[4pt] \tfrac{1}{2} + \dfrac{y - L_2}{\rho}, & L_2 \le y \le L_2 + \tfrac{1}{2}\rho, \\[4pt] 1, & y > L_2 + \tfrac{1}{2}\rho, \end{cases}$$

where, for brevity, $L_j = L_j(x)$ and $\rho = \rho(x)$.

To couple $Y(1)$ and $Y(0)$ while preserving the marginals above, we draw $(U_1, U_0)$ from a Gumbel copula $C_\theta$ with $\theta = 5$ (dimension 2), and set

$$Y(1) = F_{1|X}^{-1}(U_1), \qquad Y(0) = F_{0|X}^{-1}(U_0).$$

Finally, the observed outcome is $Y = A\, Y(1) + (1 - A)\, Y(0)$.

---

[2]Data available here: https://www.nber.org/programs-projects/projects-and-centers/oregon-health-insurance-experiment

**Quantifying margin violations.** The c.d.f. difference $D_x(y) = F_{1|x}(y) - F_{0|x}(y)$ again admits two plateaus attaining the supremum and infemum, thus violating the margin assumption for both lower and upper bound.

For $0 \leq y \leq \frac{1}{2}\rho - \frac{1}{4}g$, the left plateau is

$$D_x(y) \;=\; \frac{y}{\rho} - \frac{y - L_1}{\rho} \;=\; \frac{L_1}{\rho} \;=\; -\frac{g}{4\,\rho},$$

with an associated width of $W_L(x) = \frac{\rho}{2} - \frac{g}{4}$.

For $\frac{1}{2}\rho + \frac{1}{4}g \leq y \leq \rho$, the right plateau is

$$D_x(y) \;=\; \frac{y}{\rho} - \left(\frac{1}{2} + \frac{y - L_2}{\rho}\right) \;=\; \frac{L_2 - \frac{1}{2}\rho}{\rho} \;=\; +\frac{g}{4\,\rho}.$$

with a width of $W_R(x) = \frac{\rho}{2} - \frac{g}{4}$. The average widths are given via

$$\mathbb{E}\big[W_L(X)\big] = \mathbb{E}\big[W_R(X)\big] = \mathbb{E}\Big[\big(\tfrac{1}{2}\rho_0 + \tfrac{1}{4}\big(m(X)^2 - m(X)^3\big)\big)_+\Big].$$

Hence, the widths of the plateaus and thus also the degrees of margin violations increasing in $\rho_0$.

# I ADDITIONAL EXPERIMENTS

## I.1 (SEMI)-SYNTHETIC DATA

Here, we report full results for the synthetic and semi-synthetic datasets from Appendix G and Appendix H for a range of parameters $\gamma$ (synthetic) and $\rho$ (semi-synthetic) to complement the results from Table 2. Recall that increasing $\gamma$ *decreases* margin violation, while higher $\rho$ *increase* margin violation. The results for all methods are bound types are reported in Fig.3. For both datasets, larger margin violation leads to a larger gap between our method and the baselines. Importantly, **our method consistently outperforms the baselines**.

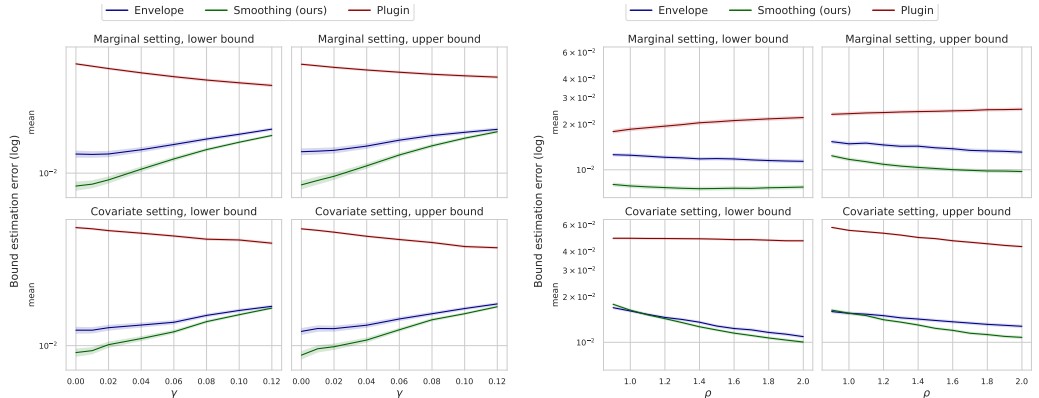

Figure 3: Results for synthetic and semi-synthetic data over a range of parameters quantifying margin violation.

## I.2 TWINS DATASET

Here, we use the binary outcome TWINS dataset from Louizos et al. (2017). The covariates $X$ are measurements and demographic attributes of various twin pairs below 2000 grams. The treatment $A$ is being born as the heavier twin, and the outcome $Y$ denotes infant mortality within the first year of life. As both counterfactual outcomes are observed, we can estimate the ground-truth treatment effect c.d.f. , which we plot along the estimated Makarov bounds in Fig.4. We see that the estimated bounds are tight and cover the ground-truth treatment effect c.d.f. , thus confirming the validity of our estimator.

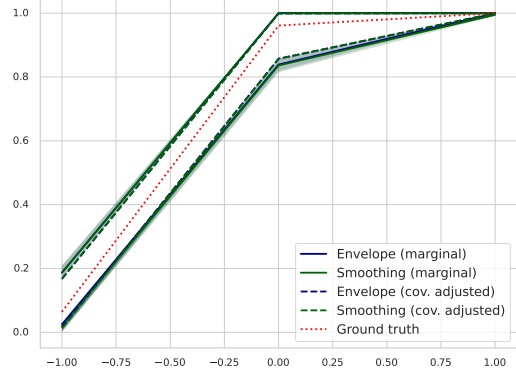

Figure 4: Estimated bounds and treatment effect c.d.f. for the TWINS dataset.

## I.3 ADDITIONAL RESULTS FOR CONSUMER COMPANY A/B TESTS

In Figure 5, we compare both the results of our smoothed estimator with the results of the envelope baseline. The upper Makarov bound estimates mostly coincide, however, our estimator is more conservative and estimates a smaller lower bound than the baseline. As our estimator is obtained by

minimizing MSE over various smoothing parameters (including the special case of large smoothing parameters coinciding with the envelope), this indicates that the ground-truth Makarov bound is smaller than the envelope estimate. In particular, this may indicate envelope estimator is too confident and may undercover the ground-truth treatment effect c.d.f. , leading to potentially wrong conclusions.

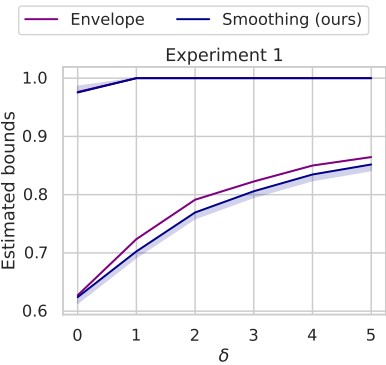

Figure 5: Estimated bounds and confidence intervals for Experiment 1.

