# OpenReview forum: "Assumption-lean inference on treatment effect distributions"
_ICLR.cc/2026/Conference — Submitted to ICLR 2026_

### Official Review · Reviewer_eg7p · 2025-10-30

**Soundness:** 2
**Presentation:** 2
**Contribution:** 2
**Rating:** 4
**Confidence:** 3

**Summary:**

na

**Strengths:**

pros:
This paper makes a clear and original contribution to causal inference by proposing an assumption-lean framework for inferring the distribution of treatment effects. Unlike prior methods that rely on restrictive margin or smoothness assumptions, the authors introduce smoothed Makarov bounds that allow valid semiparametric inference even when standard assumptions fail. The method combines theoretical rigor—through efficiency theory and bias control—with strong empirical validation on synthetic, semi-synthetic, and real-world A/B test data. Its ability to uncover heterogeneous and potentially harmful treatment effects, even when the average treatment effect is positive, highlights significant practical value for risk-aware decision-making in experiments.

con:
The main limitations lie in computational complexity and scope. The proposed smoothing-based estimators require intensive numerical integration and careful tuning of smoothing parameters, which may limit scalability in high-dimensional or large-scale settings. In addition, the framework currently applies only to binary treatments and assumes bounded outcomes, leaving open challenges for extending it to multi-valued or continuous treatments and heavy-tailed outcomes.
Despite these constraints, the paper's methodological innovation and practical relevance make it a strong step toward more robust and distributional approaches to causal inference.

**Weaknesses:**

na

**Questions:**

na

---

> ### Author Response · Authors · 2025-11-25
> **Rebuttal by authors**
>
> Thank you for your review and for giving us the opportunity for clarification. Below, we address your concerns in detail. We incorporated all points marked with **Action** into the revised version of our paper (marked in $\color{blue}blue$).
>
>
> ## Response to “Con”:
>
>
>
> * **Computational cost:** We would like to clarify that, in practice, **our method is not more computationally expensive than established alternatives:** All computations (e.g., numerical integration and smoothing parameter tuning) are done **after the nuisance model training** and thus **only require inference time of the trained nuisance models**. In our experiments, the most expensive part was training the nuisance models, which is a standard practice in causal inference and not unique to our method. Compared to nuisance model training, the additional operations for Makarov bound estimation (numerical integration, tuning of smoothing parameters) are negligible for large datasets. Furthermore, computations for multiple values of $\delta$ can be parallelized for additional speedup.
> * **Extension towards multivalued treatments:** Thanks for bringing this up. In fact, **our proposed method is readily applicable to settings with multivalued treatments.** For this, a practitioner has to specify the two treatment values $a_1$ and $a_0$ that require comparison and for which we would like to infer the ITE distribution of $Y(a_1) - Y(a_0)$. Then, the only changes to our methodology required are to replace $A$ and $1 - A$ with binary indicators $I(A = a_1)$ and $I(A = a_0)$, and the propensity contributions $\pi(X)$ and $1 - \pi(X)$ with $\pi_{a_1}(X)$ and $\pi_{a_0}(X)$ in Theorem 4.2., where $\pi_{a}(X) = P(A=a | X)$. **Action**: We made this explicit and **added an extension to multivalued treatments to our new Appendix E**.
> * **Bounded outcomes:** You are correct that our method requires bounded outcomes. However, this is not a strong practical restriction, and boundedness assumptions are standard in the causal inference literature [e.g., 1, 2, 3]. In practice, most applications yield bounded outcomes, such as the number of clicks within a specific time period for online experimentation, or patient survival, heart rate, or blood pressure in medical applications. **Action:** We acknowledge that extending our methodology to unbounded presents an interesting path for possible future research and add a discussion to our paper (**new Sec. 6**).
>
>
> ## References
>
> [1] Kallus et al. (2022). What's the Harm? Sharp Bounds on the Fraction Negatively Affected by Treatment. NeurIPS.
>
> [2] Curth et al. (2021). Nonparametric estimation of heterogeneous treatment effects: From theory to learning algorithms. AISTATS.
>
> [3] Kennedy et al. (2023). Towards optimal doubly robust estimation of heterogeneous causal effects. Electronic Journal of Statistics.

---

### Official Review · Reviewer_ZPtF · 2025-10-30

**Soundness:** 3
**Presentation:** 3
**Contribution:** 3
**Rating:** 6
**Confidence:** 3

**Summary:**

This paper focuses on the problem of estimating the full distribution of treatment effects rather than just the average treatment effect (ATE). The authors focus on estimating the so-called Makarov bounds, which define the sharp limits on the possible treatment-effect distribution given only the observed marginal outcome distributions.

Estimating these bounds is challenging because they involve non-smooth operations and previous methods either rely on strong “margin” assumptions (which are often violated in practice), or use plug-in estimators that lack valid inference guarantees. The paper proposes an assumption-lean approach that smooths the non-differentiable components of the Makarov bounds using differentiable approximations. This smoothing enables the derivation of efficient influence functions and the construction of debiased, asymptotically normal estimators. The authors also provide an explicit bound on the bias introduced by smoothing and adjust their confidence intervals accordingly. They propose two data-driven procedures for selecting the smoothing parameters: one minimizing an empirical MSE bound and another based on a Lepski-type adaptive rule. Empirically, the method outperforms plug-in and “envelope” baselines on synthetic, semi-synthetic, and real A/B test data, especially when the margin assumption fails. The results show improved bias–variance trade-offs and more reliable inference.

**Strengths:**

1. The paper targets inference on treatment-effect distributions rather than just the mean effect. This is an important and underexplored direction for causal inference and A/B testing.

2.Their method removes the need for the restrictive margin assumption used in prior work, which is often violated in realistic settings where treatment effects are constant or nearly constant. This makes the inference procedure more robust and broadly applicable.

3.The use of smooth surrogates (log-sum-exp and softplus) to approximate the non-differentiable Makarov bounds is conceptually elegant. It may have impact on other statistical topics too. Moreover, for smoothing parameters, it introduces two adaptive, data-driven methods to select them: (i) minimizing an empirical MSE upper bound and (ii) a Lepski-type adaptive selection rule. The experiments also showed their method's better performance.

**Weaknesses:**

1. Issues on widened and potentially conservative confidence intervals. Because the method adds an explicit smoothing-bias correction term to guarantee valid coverage, the resulting confidence intervals can become conservative and potentially wider than necessary. In practice, this may dilute the practical utility of the inference especially when the true bias is small.

2. A follow up question to 1 is: this is actually the cost (from smoothing) of this paper's approach by translating the difficulty in avoiding the restrictive margin assumption to the difficulty of balancing the bias and variance. Hence, although the paper provided two data-driven ways for picking the smoothing parameters, there are no theoretical guarantees on these choices. This makes the issue not totally solved (simulation or real data results could be from an optimal search of these smoothing parameters rather than from a solid guideline). Also, data-splitting approach is also used and this will reduce the data efficiency. This problem then hinges on further investigation both theoretically and empirically. So far, it is only heuristic.

**Questions:**

Please see Weakness and the following:

1. The bias bound $b(t_1,t_2)$ is derived under compact outcome support and finite Lebesgue measure. How sensitive are your guarantees to this assumption, and can it be relaxed for unbounded or categorical outcomes?

2. The framework assumes no interference between units. Could it be adapted to clustered or networked data, where spillovers exist?

3. Due to this increased confidence intervals by smoothing, do you have empirical evidence or calibration plots showing how often they over-cover relative to the nominal level? To be precise, any evaluations on whether this bias-correction term $b(t_1,t_2)$ makes confidence intervals conservative?

**Details Of Ethics Concerns:**

None.

---

> ### Author Response · Authors · 2025-11-25
> **Rebuttal by authors**
>
> Thank you very much for your positive review! Below, we address all your comments and suggestions. We incorporated all points marked with **Action** into the revised version of our paper (marked in $\color{blue}blue$).
>
>
> ## Response to Weaknesses
>
>
> 1. **Risk of conservative intervals.** Thanks, yes, correct that smoothing can lead to more conservative bounds in order to guarantee asymptotically valid confidence intervals even under margin violation. However, in practice, we tune the smoothing parameter (Sec. 4.3) to minimize a bias-variance tradeoff. If the true margin violation is small, it will automatically select larger smoothing parameters so that the resulting intervals will be tighter and not more conservative than existing methods.
> 2. **Guarantees for smoothing parameter selection and data splitting.** Thank you for pointing this out. In fact, there is a well-established theory available for both data splitting and smoothing parameter selection that is also applicable to our problem. For sample splitting and cross-fitting, established semiparametric theory applies and provides efficient estimation and asymptotic normality. For smoothing parameter selection, under certain assumptions, it can be shown that Lepskis' method can be (minimax) optimal [1].
>
>
> ## Response to Questions
>
>
>
> 1. **Bias-bound assumptions.** You are correct that our method requires bounded outcomes. However, this is not a strong practical restriction, and boundedness assumptions are standard in the causal inference literature [e.g., 2, 3, 4]. In practice, most applications yield bounded outcomes, such as the number of clicks within a specific time period for online experimentation, or patient survival, heart rate, or blood pressure in medical applications. **Action:** We acknowledge that extending our methodology to unbounded presents an interesting path for possible future research and add a discussion to our paper (**new Section 6**).
> 2. **Unit interference.** This is an interesting question. In its current state, our methodology assumes no unit interference, which is a standard assumption in causal inference [2, 3, 4]. However, we suspect that extensions to interference settings are possible, e.g., by assuming knowledge about the underlying interference graph or clusters. Combining existing methodology for unit interference in causal inference with our method is an interesting path for future research. **Action:** We added a discussion on this to our revised paper (**new Section 6**).
> 3. **Evaluation of conservative bias.** We agree that, in principle, adding the explicit bias-correction term $b(t_1,t_2)$ could make the confidence intervals conservative if the smoothing is chosen poorly. In our procedure, however, $(t_1,t_2)$ are not fixed arbitrarily: our tuning rule is explicitly designed to balance the variance term (estimated via the EIF) and the approximation bias captured by $b(t_1,t_2)$. For a fixed $\delta$, the half-width of our one-sided intervals is of the form $\text{half-width} \approx z_{1-\alpha/2} \cdot \frac{\widehat{\mathrm{sd}}(\Psi^\pm)}{\sqrt{n}} \;+\; b(t_1,t_2)$, so the interval width depends jointly on the EIF variance and the bias bound. Our tuning criterion minimizes an empirical upper bound on the MSE of the smoothed estimator, which directly trades off variance and squared bias. As a result, values of $(t_1,t_2)$ that would make $b(t_1,t_2)$ unnecessarily large are disfavored, and the selected smoothing level keeps approximation bias and sampling variability of comparable order, yielding intervals that are theoretically valid while avoiding unnecessarily large over-coverage in practice.
>
>
> ## References
>
> [1] Goldenshluger et al (2011). Bandwidth selection in kernel density estimation: Oracle inequalities and adaptive minimax optimality. Annals of Statistics.
>
> [2] Curth et al. (2021). Nonparametric estimation of heterogeneous treatment effects: From theory to learning algorithms. AISTATS.
>
> [3] Kennedy et al. (2023). Towards optimal doubly robust estimation of heterogeneous causal effects. Electronic Journal of Statistics.
>
> [4] Kallus et al. (2022). What's the Harm? Sharp Bounds on the Fraction Negatively Affected by Treatment. NeurIPS.

---

### Official Review · Reviewer_LnBg · 2025-11-05

**Soundness:** 2
**Presentation:** 3
**Contribution:** 2
**Rating:** 2
**Confidence:** 4

**Summary:**

The article under review proposes an efficient method for estimating bounds on the distribution of treatment effects. The approach combines two existing ideas: debiased estimators and smoothing techniques. This combination yields element-wise semiparametrically efficient estimators while simultaneously addressing the smoothing bias introduced by the technique.

**Strengths:**

The article relies on recent techniques from the literature, and the analysis appears sound.

**Weaknesses:**

1. **Bounds**

- The paper's abstract, which argues that existing workflows "overlook distributional risks," appears to be overstated. There is a considerable literature on partial identification for treatment effect distributions that directly addresses this. The more pressing challenge, which the paper does not discuss, is the practical utility of these existing methods. Often, the identified bounds—and particularly their confidence bands—are too wide to provide meaningful guidance, thus limiting their impact. This essential context is missing from the paper's framing.


- This omission becomes more concerning in light of the paper's own results. The authors are critical of existing work for its "non-standard inference methods," yet the figures presenting their own estimations omit the corresponding confidence bands.
To show the practical utility of the proposed method and maintain consistency with its own critiques, reporting these confidence bands is essential. Hiding this information prevents a full and fair evaluation of the method's practical contribution.

2. **Theoretical Contributions**

- *Concerns Regarding Novelty and Development*:
While Table 1 summarizes the paper's stated contributions, the core techniques, such as debiasing and smoothing methods, are established tools from the existing literature. This reliance on established methods raises concerns about the paper's marginal contribution and novelty. Moreover, the analysis appears underdeveloped in key areas, as detailed below.

- *On the Definition of "Efficiency"*:
The paper's claim of an "efficient" estimator appears to hold only in an element-wise sense (i.e., for each plug-in estimator). However, the primary quantity of interest is the **interval estimator**. The paper does not demonstrate that efficiency of the individual endpoints implies efficiency for the interval itself. This is a critical distinction that needs to be rigorously addressed.

- *Omission of Uniform Inference*:
Furthermore, when discussing distributions, inference should not be restricted to point-wise estimation. The more appropriate and relevant analysis would consider a **uniform bound** that holds over the entire distribution. This approach has been extensively investigated in the literature, and it is unclear why the authors did not conduct such an analysis. This omission is a significant gap, as it avoids the standard method for this class of problem.

3. **Simulation and Synthetic Data Studies**

- The table reports only the Mean Squared Error (MSE). Given that the proposed smoothing method intentionally introduces bias, bias must be reported separately. Presenting only MSE hides the trade-off at the heart of the technique.

- The MSE of each interval endpoint is of limited relevance. The primary object of interest is the interval itself. The authors should report metrics appropriate for interval estimation, such as average interval length and empirical coverage probability, rather than treating the problem as one of point-estimation.

**Questions:**

- Questions 1-3: Please see Weaknesses 1-3.

- Question 4.
The paper highlights the lack of "asymptotic normality" in existing methods as a significant drawback. This criticism, however, appears overstated. Many well-established estimators in statistics feature non-standard asymptotic distributions, and this alone does not invalidate them. This critique is particularly questionable given that the proposed method introduces its own set of practical complications, namely smoothing bias and the necessity of selecting tuning parameters. Is a more fair discussion of these respective trade-offs helpful?

- Question 5.
The paper’s asymptotic analysis lacks clarity, particularly regarding the convergence rate and the precise role of the smoothing bias in the large-sample results.
While Corollary 4.3 provides a formula for a confidence interval, the main text lacks the explicit weak convergence results necessary to formally justify this corollary.
Furthermore, the proposed confidence interval depends on numerous nuisance parameters and functions. The paper fails to demonstrate that the interval remains theoretically valid when these nuisance components are estimated via a "plug-in" approach. This is a critical omission, as the validity of such a procedure is not self-evident. This concern is compounded by the vague specification of the paper's underlying assumptions, which makes it impossible to verify the method's theoretical soundness.
Where can I found those information in the paper?

**Details Of Ethics Concerns:**

Not relevant for the article.

---

> ### Author Response · Authors · 2025-11-25
> **Rebuttal by authors #1**
>
> Thank you very much for your constructive review! Below, we address your comments and suggestions in detail. We incorporated all points marked with **Action** into the revised version of our paper (marked in $\color{blue}blue$).
>
> ## Response to Weaknesses
>
> 1. **Bounds.**
>     * **Framing in Abstract.** Thank you for your feedback. In our abstract, we aimed to motivate the estimation of partial identification bounds via practical examples. However, we acknowledge that the main scope of our paper is not the derivation of partial identification bounds but the estimation of said bounds. **Action:** We changed our abstract and now more clearly frame the scope of our paper and emphasize our contribution of proposing a new bound estimator.
>     * **Confidence intervals in experiments.** Please allow us to clarify this misunderstanding. **We report confidence intervals for all our experimental results.** In Figure 2, we report the estimated Makarov bounds (both lower and upper) and their corresponding confidence intervals that are computed via Corollary 4.3. The confidence intervals are reported as shaded areas around the point estimates. Note that these intervals are quite small due to the large sample size used, so that, on a first glance, it may appear as if no intervals are reported.
> 2. **Theoretical contributions.**
>     * **Novelty.** Thank you for allowing us to elaborate on our contributions. Our main contribution is the derivation of an efficient estimator based on smoothing. While you are correct that key techniques are known in the literature and have been applied to other causal inference problems, to our knowledge, we are the first to specifically adapt them for Makarov bounds. Our technical contributions **include both the derivation of the smoothed bounds (Lemma 4.3) and the derivation of the efficient estimator** (based on the EIF) under smoothing (Theorem 4.2). Hence, we are consistent with well-established causal inference contributions that proposed EIF-based estimators for previously overlooked causal estimands [e.g., 1, 2, 3, 4].
>     * **Definition of efficiency for intervals.** You are correct that our bound estimators are efficient for the interval endpoints (lower and upper Makarov bounds). Please note that any interval is defined precisely by its boundary points, which implies that **any pair of efficient endpoint estimators for the boundary points is an efficient estimator for an interval.** In fact, this approach to interval estimation is standard in the literature on semiparametric statistics and has been applied to various interval estimation problems in e.g., partial identification [e.g., 1, 2, 4].
>     * **Uniform inference.** Thank you for bringing up this important point. **Action:** We took your feedback to heart and **extended our methodology to provide uniform confidence intervals** over a set of different evaluation points $\delta$. For this, we propose a multiplier bootstrap method, which we write out in our **new Appendix D**.
> 3. **Simulation and synthetic data studies.** Thank you for highlighting both the role of bias and the importance of interval-level performance. Our theoretical framework is explicitly built around a bias--variance trade-off for the estimators of the Makarov bounds: Eq. (14) decomposes the MSE of $\hat\rho^\pm_{t_1,t_2}(\delta)$ into a squared smoothing-bias term plus a variance term driven by the EIF, and the smoothing bias is controlled analytically via the bound $b(t_1,t_2)$ in Lemma 1. The use of MSE in Table 1 is therefore intended to summarize the *net* effect of this trade-off in the metric that follows directly from our theory, rather than to obscure the presence of bias. At the same time, we agree that the ultimate inferential object is the interval for $\rho(\delta)$. Corollary 1 constructs one-sided confidence intervals whose endpoints take the form $\hat\rho^\pm_{t_1,t_2}(\delta)\;\pm\;\text{(standard error from the EIF)}\;\pm\;b(t_1,t_2)$, so improvements in the MSE of $\hat\rho^\pm_{t_1,t_2}(\delta)$ translate directly into tighter intervals at a fixed nominal level, while the explicit addition of $b(t_1,t_2)$ ensures robustness to smoothing bias and favors valid coverage. Thus, our empirical comparisons are diagnostics for the quality of the interval endpoints, not an attempt to recast the problem as pure point estimation.

---

> ### Author Response · Authors · 2025-11-25
> **Rebuttal by authors #2**
>
> ## Response to Questions
>
>
> * **Question 4 (asymptotic normality and trade-offs).** Thank you for raising this point. We agree that non-standard asymptotics are common in statistics, and that the lack of asymptotic normality alone does not invalidate an estimator. Our intention was not to suggest that existing plug-in or envelope estimators are unusable, but rather to highlight a *practical* limitation: in realistic A/B testing settings, it is difficult to obtain simple, provably valid confidence intervals when (i) the asymptotic distribution is unknown, or (ii) standard asymptotics rely on strong assumptions such as the margin assumption.
> * **Question 5 (Formal assumptions and weak convergence.)** Thank you for pointing this out. **Action:** We reworked the proof of our Corollary 4.3 (in Appendix A.3) and now formally state all assumptions and convergence results. In particular, we require standard rate conditions on the nuisance estimators $\|\hat{\eta}-\eta\|_{2} = o_p(n^{-1/4})$ as well as sample splitting to establish weak convergence (see e.g., [5]).
>
>
> ## References
>
> [1] Kallus et al. (2022). What's the Harm? Sharp Bounds on the Fraction Negatively Affected by Treatment. NeurIPS.
>
> [2] Lewis et al. (2023). Covariate-assisted bounds on causal effects with instrumental variables. Journal of the Royal Statistical Society Series B.
>
> [3] Curth et al. (2021). Nonparametric estimation of heterogeneous treatment effects: From theory to learning algorithms. AISTATS.
>
> [4] Dorn et al. (2025). Doubly-Valid/Doubly-Sharp Sensitivity Analysis for Causal Inference with Unmeasured Confounding. Journal of the American Statistical Association.
>
> [5] Chernozhukov et al. (2018). Double/debiased machine learning for treatment and structural parameters. The Econometrics Journal.

---

### Official Review · Reviewer_Dk4G · 2025-11-05

**Soundness:** 3
**Presentation:** 2
**Contribution:** 1
**Rating:** 2
**Confidence:** 5

**Summary:**

This paper develops a new way to estimate Makarov bounds on treatment effect distributions by replacing the non-smooth max/min operators with smooth log-sum-exp (LSE) approximations. This smoothing enables valid inference even when traditional semiparametric estimators fail due to non-unique extrema (margin violations).

**Strengths:**

1. The idea of using log-sum-exp smoothing to recover differentiability is straightforward and simple.
2. The paper provides a principled and theoretically grounded framework for valid semiparametric inference under margin violations.

Overall, the work is technically sound, and well supported by strong theoretical guarantees and empirical validation.

**Weaknesses:**

_1. Weak litearture review._
The paper omits a substantial body of work on Quantile Treatment Effects (QTE), which  serves as a core approach to distributional causal inference. This omission weakens the positioning of the proposed Makarov-based framework within the broader context of distributional effect estimation. I recommend to discuss prior QTE literature, and clearly articulate why the Makarov-based approach is advantageous when the joint distribution of potential outcomes is unidentified, whereas QTE only captures marginal contrasts.

_2. Unclear motivation and significance._
The paper does not sufficiently justify why margin violations pose a critical practical problem. While the theory is correct, the empirical motivation could more clearly demonstrate concrete failure cases of existing methods under margin violations, ideally with a real-world example rather than synthetic illustrations. Without such evidence, the significance of the proposed smoothing appears unclear.

_3. Incremental contribution_
The main methodological idea (replacing non-smooth max/min operators with log-sum-exp smoothing) is well-known in optimization and statistical theory. The paper mainly applies this existing trick to the Makarov bounds without offering fundamentally new theoretical insight or stronger guarantees beyond standard smoothing arguments.

_4. Presentation_
The table is too packed and the fonts are too small. I believe this is a violation of the font-size regulation ("do not change font sizes")

_5. Title and scope._
The title “Assumption-Lean Inference on Treatment Effect Distributions” is overly broad relative to the actual methodological contribution. The paper’s method still depends on very strong assumptions (e.g., ignorability condition, discrete treatments, overlap, etc.) so the “assumption-lean” claim seems overstated. A more concrete title explicitly reflecting the smoothing-based inference for Makarov bounds convey the scope and focus.

**Questions:**

Q1. How tight or sharp the smooth approximation of the Makarov bound?

---

> ### Author Response · Authors · 2025-11-25
> **Rebuttal by authors**
>
> Thank you for your constructive review and your helpful comments. Below, we have drafted careful responses to your suggestions. We incorporated all points marked with **Action** into the revised version of our paper (marked in $\color{blue}blue$).
>
>
> ## Response to Weaknesses
>
> 1. **Missing literature review on quantile treatment effects (QTEs).** Thank you for pointing this out. **Action:** We followed your suggestion and included a new paragraph reviewing QTEs in our related work [e.g., 1, 2]. We also emphasize that this literature stream is **different** from our setting, as it focuses on contrasts of quantiles $q_\alpha(Y(1)) - q_\alpha(Y(0))$, while we are interested in the distribution/ quantiles of the individual treatment effect $Y(1) - Y(0)$. Note that QTEs do not quantify quantiles of the treatment effect distribution: QTEs describe how treatment shifts the marginal distributions of potential outcomes across individuals (for example, how the $\alpha$-quantile under treatment differs from the $\alpha$-quantile under control), while the treatment effect distribution would characterize the distribution of individual-level causal effects $Y(1)-Y(0)$ in the population. As you mentioned correctly, the key difference is that the treatment effect distribution is **only partially identified via Makarov bounds**, while QTEs are point identified under standard assumptions.
> 2. **Margin violation in real-world datasets.** Margin assumption violations are common in practical causal inference problems and can arise even in simple settings with partially constant treatment effects (as in our example in Fig. 1). Importantly, such (near-) constant effects are not just a feature of stylized toy examples. They naturally occur when outcomes are discrete or exhibit zero inflation, which is a common occurrence in applied work. For instance, in A/B tests with binary or highly skewed outcomes such as clicks, conversions, or purchases (many units with exactly zero events). In these cases, margin assumptions are typically not satisfied, making a smoothing-based approach particularly relevant. **Action**: We sharpened this motivation in the revised version of our paper.
> 3. **Contribution.** Thank you for allowing us to elaborate on our contribution. Our main contribution is the derivation of an efficient estimator based on smoothing. While you are correct that smoothing-based approaches have been proposed for other causal inference problems, to our knowledge, we are the first to specifically adapt them for Makarov bounds (Lemma 4.3). Furthermore, our technical contributions **do not only include the proposed smoothing approach, but also the derivation of the efficient estimator** (based on the EIF) under smoothing (Theorem 4.2). Therein, we are consistent with well-established causal inference contributions that proposed EIF-based estimators for previously overlooked causal estimands [e.g., 3, 4, 5]. Additionally, we also propose methods for smoothing parameter selection based on an upper bias bound on the smoothing approximation error.
> 4. **Small table fonts.** Thank you for pointing this out. **Action**: We followed your suggestion and increased the font and table sizes.
> 5. **Title and scope.** Thank you for the feedback. We acknowledge that the current title may be too broad and are open to renaming the paper. Following your suggestion, a possible title may be “Smoothing-based inference for Makarov bounds on the individual treatment effect”. We are open to refining this during the discussion phase.
>
>
> ## Response to Questions
>
> * Q1: **Tightness of smooth approximation**. The tightness of the smooth bound approximation generally depends on the choice of the smoothing parameters $t_1$ and $t_2$ and the volume of the outcome space $|\mathcal{Y}|$. It is upper bounded by the term $b(t_1, t_2)$ from our Lemma 4.1. In practice, choosing the right smoothing parameters corresponds to a tradeoff between smoothing bias and estimation variance as the asymptotic variance based on the EIF depends on both $t_1$ and $t_2$. We propose two methods for automatic smoothing parameter selection in Section 4.3.
>
>
> ## References
>
> [1] Chernozhukov et al. (2004). An IV Model of Quantile Treatment Effects. Econometrica.
>
> [2] Firpo et al. (2007). Efficient Semiparametric Estimation of Quantile Treatment Effects. Econometrica.
>
> [3] Kallus et al. (2022). What's the Harm? Sharp Bounds on the Fraction Negatively Affected by Treatment. NeurIPS.
>
> [4] Lewis et al. (2023). Covariate-assisted bounds on causal effects with instrumental variables. Journal of the Royal Statistical Society Series B.
>
> [5] Curth et al. (2021). Nonparametric estimation of heterogeneous treatment effects: From theory to learning algorithms. AISTATS.

---

### Author Response · Authors · 2025-11-25
**Response to all reviewers**

Thank you very much for the constructive evaluation of our paper and your helpful feedback! We sincerely appreciate the time and effort you have taken to review our paper. We have addressed your comments in the individual responses below and also uploaded a revised version of our paper, highlighting key changes in $\color{blue}blue$.

We will incorporate all changes into the camera-ready version of our paper. Given these improvements, we are confident that our paper will be a valuable contribution to the causal machine learning literature.

---

### Meta-Review · Area_Chair_d4Uk · 2026-01-14

**Summary:**

This paper provides a smooth surrogate for Markov bound, which is used on cumulative distribution of treatment effects and provides an optimal estimation of smoothing parameters. The reviewers expressed several concerns including lack of novelty and contributions.
I think the implications of some technical results are not clear. For example, in theorem 4.2, it is not clear to me what the high level take-aways. I think some deeper explanation of the theoretical results are needed. Tables in the experimental sections should better presented. At the current state, they are not readable. Entries in Table 3 are too small. In Figure 3, the Y axis and X axis labels are too small too read. Reviewer Dk4G also pointed out poor literature review. Also, the title of the paper is not very suitable with the main idea. I think the paper should be rejected.

**Reviewer Concerns:**

Authors did their best effort to address the reviewer concerns. But many concerns require complete revamping of the current work.

**Reviewer Scores:**

I do not think any reviewer would have changed their score.

---

### Decision · Program_Chairs · 2026-01-26

Reject